# Interpretable Bilingual Multimodal Large Language Model for Diverse Biomedical Tasks

**Lehan Wang**[1], **Haonan Wang**[1], **Honglong Yang**[1], **Jiaji Mao**[2], **Zehong Yang**[2],
**Jun Shen**[2], **Xiaomeng Li**[1]*

## Abstract

Several medical Multimodal Large Languange Models (MLLMs) have been developed to address tasks involving visual images with textual instructions across various medical modalities, achieving impressive results. Most current medical generalist models are region-agnostic, treating the entire image as a holistic representation. However, they struggle to identify which specific regions they are focusing on when generating a sentence. To mimic the behavior of doctors, who typically begin by reviewing the entire image before concentrating on specific regions for a thorough evaluation, we aim to enhance the capability of medical MLLMs in understanding anatomical regions within entire medical scans. To achieve it, we first formulate **Region-Centric tasks** and construct a **large-scale dataset, MedRegInstruct,** to incorporate regional information into training. Combining our collected dataset with other medical multimodal corpora for training, we propose a **Region-Aware medical MLLM, MedRegA**, which is the first bilingual generalist medical AI system to simultaneously handle image-level and region-level medical vision-language tasks across a broad range of modalities. Our MedRegA not only enables three region-centric tasks, but also achieves the best performance for visual question answering, report generation and medical image classification over 8 modalities, showcasing significant versatility. Experiments demonstrate that our model can not only accomplish powerful performance across various medical vision-language tasks in bilingual settings, but also recognize and detect structures in multimodal medical scans, boosting the interpretability and user interactivity of medical MLLMs. Our project page is https://medrega.github.io.

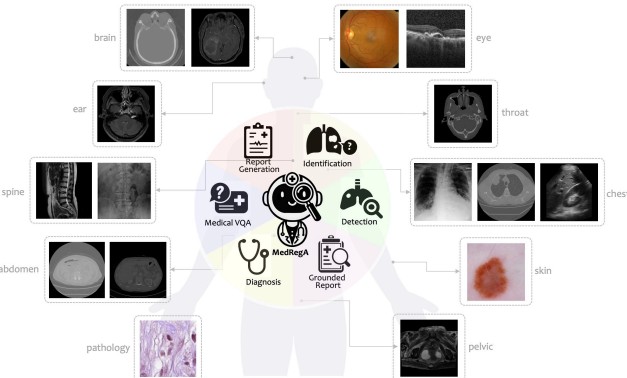

Figure 1: **MedRegA**, an interpretable bilingual generalist model for diverse biomedical tasks, represented by its outstanding ability to leverage regional information. MedRegA can perceive 8 modalities covering almost all the body parts, showcasing significant versatility.

---
*Corresponding to Xiaomeng Li (eexmli@ust.hk). [1]The Hong Kong University of Science and Technology. [2]Sun Yat-Sen Memorial Hospital, Sun Yat-Sen University.

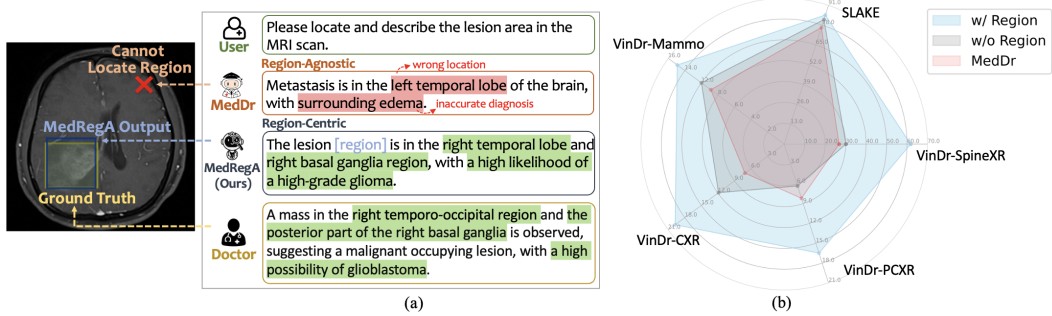

Figure 2: **The significance of Region-Centric ability.** (a) Comparison between the region-agnostic model (MedDr) and the region-centric MedRegA in analyzing lesion area within the medical scan. (b) Performance comparison of prompting the model with and without regional information on five benchmarks of Visual Question Answering (VQA) and classification tasks.

# 1 INTRODUCTION

In recent years, Multimodal Large Language Models have witnessed rapid advancements (OpenAI, 2023; Liu et al., 2024; Reid et al., 2024; Bai et al., 2023; Chen et al., 2024b),which also appear promising for adaption into the healthcare field. By integrating visual and textual modalities, MLLMs can address diverse medical needs, such as patient consultations, medical report generation, and disease diagnosis. Several medical Artificial Intelligence (AI) models have been proposed to tackle different tasks involving visual images with textual instructions across multiple medical modalities and achieved impressive results (Moor et al., 2023; Li et al., 2024; Tu et al., 2024; Wu et al., 2023; He et al., 2024; Zhang et al., 2024a). However, most current medical generalist models are **region-agnostic**, tailored to treat the entire image as a holistic representation, but struggling to detect or reason about specific regions within a medical scan. This lack of spatial awareness often leads to inaccurate descriptions that negatively affect the quality of the generated reports. As illustrated in Figure 2 (a), region-agnostic models, such as MedDr (He et al., 2024), mistakenly identified the lesion area in the left part of the brain, while the ground truth is located in the right part. This incorrect localization led MedDr to classify the lesion as surrounding edema instead of glioma, as radiologists noted. Consequently, this makes it difficult for clinicians to verify, trust, or revise the generated report, thereby hindering the model's interpretability and clinical usability.

The primary challenge arises because previous methods were not trained on **region-centric tasks**. Specifically, medical reports often consist of descriptions of multiple regions, making it unclear which regions the model is addressing for each generated sentence. This ambiguity reduces reliability and does not align with the clinical workflow. Radiologists typically begin by reviewing the entire image and then concentrating on specific regions to ensure a thorough evaluation. Inspired by this process, we aim to integrate region-centric information into medical MLLMs, mirroring the radiologist's workflow. This approach is crucial as it ensures the model focuses on the correct regions being described, whether normal, abnormal, uncertain, or in any other state.

To address the above limitations, we first introduce three **Region-Centric tasks**: *(1) Region-to-Text Identification* aims to identify structures, organs, or any abnormalities within a given bounding box region as input; *(2) Text-to-Region Detection* aims to accurately locate the positions of structures or abnormalities described in the instruction by providing bounding boxes; *(3) Grounded Report Generation* aims to generate detailed reports with corresponding bounding boxes for associated anatomies in the medical image. To enable MLLMs to perform these three tasks, we constructed a large-scale dataset, **MedRegInstruct**, encompassing all the above tasks. We collect real clinical data from Sun Yat-Sen Memorial Hospital, including approximately 25K Chinese scan-report pairs of X-ray, CT, and MRI modalities from 15K patients. Then, we propose an automatic labeling system to curate grounded reports, lowering the expense of manually annotating fine-grained organs within each medical scan.

Combining our collected dataset MedRegInstruct with other medical multimodal corpora for training, we propose a **Region-Aware medical MLLM**, **MedRegA**, which is the first bilingual general-

ist medical AI system to simultaneously handle image-level and region-level vision-language tasks across a broad range of modalities, such as radiology, pathology, dermatology, and ophthalmology; see Figure 1. Experiments presented in Figure 2 (b) show that our MedRegA can precisely output the bounding box of the specific region being focused on, while other existing best-performing medical MLLMs cannot. This significantly helps the model to correctly describe the condition of certain areas (Figure 2-a) and further enhances other downstream tasks such as visual question answering and diagnosis (Figure 2-b). In summary, our MedRegA outperforms MedDr by 3.91% and 8.03% (BLEU-1) on MIMIC-CXR and IU-Xray in English report generation, with an additional improvement of 27.34% for generating Chinese reports. Besides, our MedRegA not only enables three region-centric tasks but also achieves the best performance for visual question answering, report generation, and medical image classification over 8 modalities, showcasing significant versatility.

To summarize, our contributions can be concluded as follows:

(1) We establish **Region-Centric tasks** with a large-scale dataset, **MedRegInstruct**, where each sample is paired with coordinates of body structures or lesions. This would expand the model's functionality to perceive regions within the medical scan, encouraging the model to focus on critical areas and improving interpretability. To benchmark the regional ability of medical MLLMs in performing those tasks, we propose a **Region-Aligned evaluation** framework to assess the quality and alignment of the output text and regions.

(2) Based on the proposed dataset, we develop a **Region-Aware medical MLLM**, **MedRegA**, as a bilingual generalist medical AI system to perform both image-level and region-level medical vision-language tasks, demonstrating impressive versatility. In the inference stage, we introduce **Regional CoT** (Chain-of-Thought) to further enhance model performance with its spatial knowledge.

(3) We evaluate our model on comprehensive medical tasks, including visual question answering, report generation, image classification, region identification, region detection, and grounded report generation. MedRegA outperforms SOTA methods by a large margin on traditional visual-language and region-centric tasks.

## 2 RELATED WORK

**Vision-Language Datasets for Medicine**. In the advancement of Multimodal Large Language Models, there is a strong demand for image-text datasets that serve as pretraining data sources of MLLMs, as these datasets play a crucial role in impacting model performance. The most universally applied vision-language datasets in medicine including those for visual question answering (VQA) tasks (Lau et al., 2018; Liu et al., 2021; He et al., 2020; Zhang et al., 2023a; Chen et al., 2024a) and medical report generation (MRG) tasks (Johnson et al., 2019; Hamamci et al., 2024). However, these datasets lack region annotations relevant to the texts, impeding MLLMs from further understanding structural information within medical scans. Huang et al. (2024); Xie et al. (2024) has attempted to close the gap by incorporating region grounding into multimodal conversations, but had limited access to manually written descriptions from real clinical scenarios. Although works of Zhang et al. (2024b); Lei et al. (2024) constructed region-wise reports paired with anomaly masks from clinical data, they concentrated on chest CT and brain MRI respectively, still leaving a blank space for other body structures such as colon, liver, and pancreas. As a consequence, existing datasets have not met the requirements in developing a generalist MLLM in healthcare, especially to comprehend regional information and accommodate diverse multimodal inputs from a wide range of medical systems. This poses significant challenges in creating a model that can effectively assist clinicians across various specialties.

**Medical Multimodal Large Language Models**. The rapid progression in Multimodal Large Language Models has driven the integration of MLLMs into the medical field to enhance diagnostic processes, medical research, and patient care. The development of large-scale medical models follows the trend of progressively scaling up the application scope. Several AI assistants are specifically designed for a single modality. Given the abundant public datasets for Chest X-ray (CXR) images, Thawkar et al. (2023); Pellegrini et al. (2023); Chen et al. (2024c); Hyland et al. (2023) have facilitated MLLMs' ability to understand CXR scans. Specifically for pathology, Lu et al. (2023) curated a large pathological image dataset and presented a pathological AI assistant. Since the large-scale parameters of MLLMs enable them to store knowledge across various modalities,

extensive works enlarge the coverage of medical imaging to deploy more general and applicable medical MLLMs (Wu et al., 2023; Chaves et al., 2024; Bai et al., 2024; Zhang et al., 2023a; Moor et al., 2023; Li et al., 2024; Liu et al., 2023; Tu et al., 2024; He et al., 2024; Zhang et al., 2024a). However, without the capability of recognizing and detecting anatomies within medical scans, previous models are still confronted with challenges in providing expert-like detailed descriptions of specific structures or lesions, which notably limits their reliability and interpertability.

**Multimodal Large Language Models with Regional Ability**. Enhancing the regional ability to perceive particular objects and ground responses with related areas is crucial for facilitating visual understanding, making it a prevailing target in training MLLMs for natural images (Wang et al., 2023; Peng et al., 2023; Guo et al., 2024). However, enabling MLLMs to detect and ground fine-grained structures in medical scans remains largely underexplored. Current research primarily focused on a limited set of region-related tasks involving only a few modalities. For instance, models proposed by Huang et al. (2024); Alkhaldi et al. (2024); Zhou et al. (2024) are capable of detecting single structures based on the given prompts, but failing to delineate fine-grained descriptions for each region, as a clinician could. Bannur et al. (2024) introduced a grounded report generation task and developed an MLLM to generate descriptions of all findings in a medical image accompanied with corresponding location, but the model was restricted to receiving merely chest X-ray input. Based on these limitations, we aimed to fully integrate the function of recognizing, detecting and describing regions for a wide range of modalities into our proposed model.

## 3 REGION-CENTRIC DATASET CONSTRUCTION

In this section, we introduce the MedRegInstruct dataset, which is composed of two subsets: (1) the Region-Text Dataset containing 550K triplets of image, question-answer pair, and corresponding regions, (2) the Region-Grounded Dataset including 240K instances where each image combines a report annotated with regions and their fine-grained descriptions. Data distribution details can be found in Appendix A.1. Compared with the previous vision-language datasets for medicine (Liu et al., 2021; Johnson et al., 2019; Hamamci et al., 2024), our dataset not only places an extra emphasis on cultivating the regional abilities of medical MLLMs, but also substantially enhances diversity by integrating a wider range of anatomical structures and a bilingual setting for report generation.

### 3.1 REGION-TEXT DATASET

The creation of our Region-Text dataset stems from SA-Med2D-20M (Ye et al., 2023), which is a large-scale segmentation dataset containing 2D medical images of almost all the body parts. We filter approximately 285K images from the original dataset to maintain the diversity of anatomical structures and anomaly lesions. The original labeled segmentation masks are converted into bounding boxes to represent the highlight region. The image-region pairs were formulated into two forms: (1) *Region-to-Text*: the model outputs the information of a highlighted region given the bounding box; (2) *Text-to-Region*: the model locates the specific regions by generating bounding boxes. To obtain question-answer pairs, we fill the predefined templates with region names and coordinates. In statistics, the Region-Text dataset contains 550K data items, with approximately a half designated as Region-to-Text and the other half as Text-to-Region respectively.

### 3.2 REPORT-GROUNDED DATASET

The report-grounded dataset is sourced from two databases, MIMIC-CXR dataset (Johnson et al., 2019), and our in-house clinical data containing 25K X-ray, MRI and CT scans from 15K patients. The in-house data covers multiple regions, including the brain, abdomen, chest, spine, and pelvis. In this section, we will illustrate the automatic data processing procedure to curate grounded reports following three steps: Image-Report Pair Construction, Report Refinement and Structure Detection.

**Image-Report Pair Construction.** For the MIMIC-CXR dataset, we follow previous works (Wu et al., 2023) to utilize both frontal and lateral images, and include findings and impressions in the report. For our in-house dataset, we extract central slices from each 3D scan to formulate the 2D inputs, which typically provide the most representative views of the anatomical structures. In total, we construct 95K image-report pairs from the collected clinical data, which can also be employed to stimulate the bilingual abilities of MLLM by enhancing it to generate structured reports in Chinese.

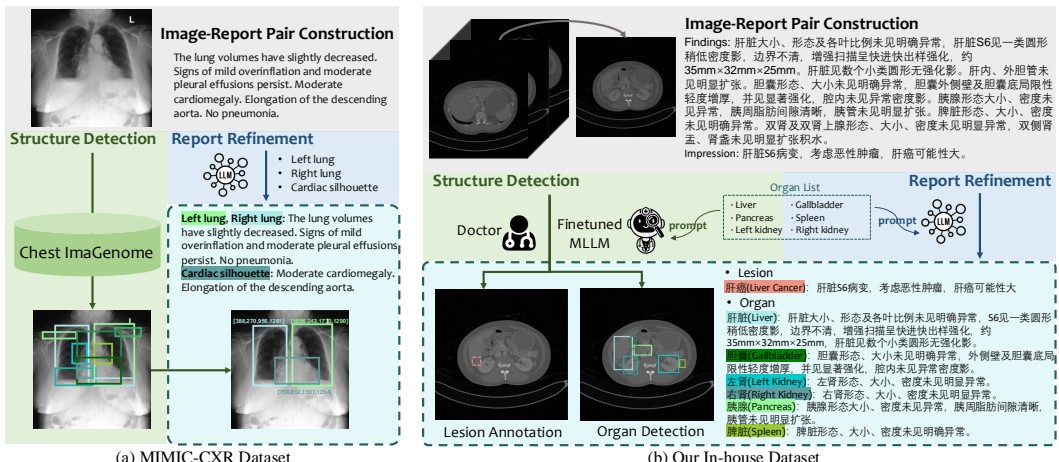

Figure 3: **Data Construction Pipeline for Report-Grounded Dataset.** The automatic data processing procedure is composed of three steps: Image-Report Pair Construction, Report Refinement, and Structure Detection.

**Report Refinement.** Our objective is to decompose raw report data into fine-grained descriptions for each organ mentioned in medical scans. We observe that the open-source InternLM model (Cai et al., 2024) demonstrates impressive textual comprehension ability for Chinese medical reports. To better leverage the LLM's information extraction ability, we first implement a rule-based strategy to summarize a coarse overview of all the organs referred in the report. Subsequently, we employed InternLM to segment each report into detailed descriptions given the organ list as the prompt.

**Structure Detection.** For the MIMIC-CXR dataset, which has been labeled with bounding box coordinates of 29 anatomical locations by Chest ImaGenome (Wu et al., 2021), we select 12 standardized ones in chest. Then, we combine the selected bounding boxes with the corresponding description in the segmented reports to form the region-grounded reports for chest X-ray scans. Overall, 220K grounded reports are constructed from the MIMIC-CXR dataset. For our in-house dataset, certain scans are manually labeled with the lesion area, such as brain tumor, lung cancer, and other abnormalities. Nevertheless, the region labels of anatomical structures are still in demand. To automatically locate the observed organs within the medical scan, we first finetune an MLLM based on the Region-Text dataset, which contains scans of the same modalites and organs as our collected data. Then, we input the collected images into the finetuned MLLM, prompted with the organs covered in the reports, in order to acquire the corresponding bounding boxes. Subsequently, we match the annotated abnormality coordinates with descriptions for the lesion part and the detected organ regions with corresponding parts in the segmented report.

**Human Validation.** To evaluate the quality of the automatically annotated part of our dataset and present a quantitative assessment, we randomly select 50 samples and asked 2 experts to create manual labels for comparison. For Report Refinement, the sentence-level accuracy is 93.33%. For Structure Detection, the accuracy of generated bounding boxes achieves 72% compared with the human annotations. We also conducted a visual evaluation and found that although most bounding boxes are slightly larger, they can still encompass the target region. This is sufficient for our approach since we only require a localization rather than a tight and accurate bounding box.

## 4 REGION-AWARE MEDICAL MLLM

The Region-Aware Medical MLLM MedRegA served as an interpretable bilingual medical AI system to tackle a variety of vision-language tasks, including visual question answering, report generation and classification, with a specialized regional ability for Region-Centric tasks. In this section, we first introduce the formulation of Region-Centric tasks in our model. Secondly, we illustrate the training process of the model. Finally, we propose a Region-Guided generation strategy aimed at further improving the inference quality.

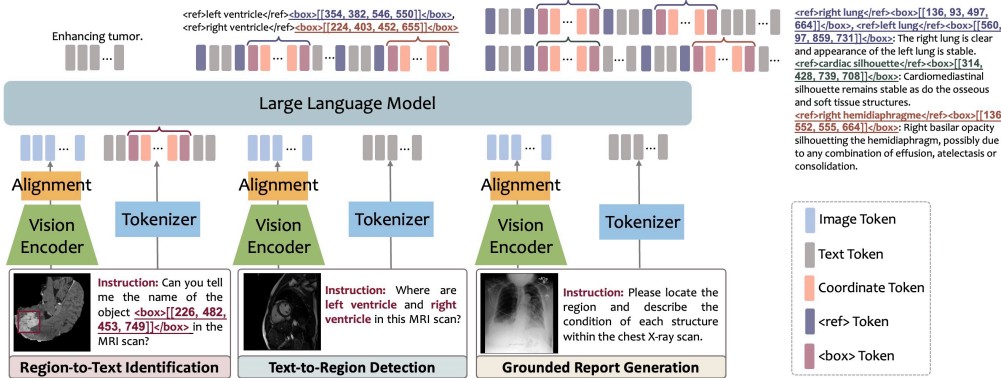

Figure 4: **An illustrative example of performing region-centric tasks with MedRegA.**

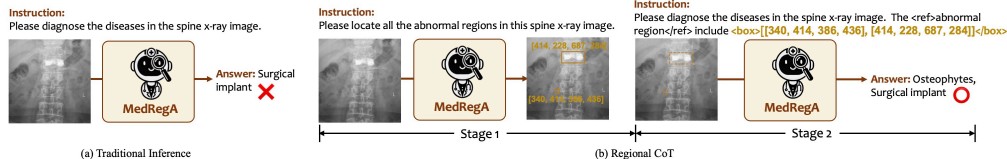

Figure 5: **Comparison of traditional inference and generation pipeline with Regional CoT.**

**Region-Centric Task Formulation**. We define the Region-Centric tasks from three perspectives: (1) Region-to-Text Identification, where the model outputs the name of the specified region; (2) Text-to-Region Detection, where the model detects the area of the given organ or anomaly; (3) Grounded Report Generation, where the model generates a report for the medical scan, aligning each description with the corresponding region. To enable the model to encode the regional information, we represented regions in the format of bounding boxes $[x_1, y_1, x_2, y_2]$, where $(x_1, y_1)$ indicate the top left point of the box and $(x_2, y_2)$ denotes the bottom right one. All the values are normalized into integers within the range of $[0, 1000)$. These bounding boxes were inserted into the sentences surrounded by a pair of special tokens, like `<box>`$[x_1, y_1, x_2, y_2]$`</box>`, while the correlated object was marked with `<ref></ref>`, as illustrated in Figure 4.

**Model Training**. To train a general medical MLLM not limited to understanding regions, we process and integrate various public datasets targeted at different tasks, including visual question answering, report generation, organ classification, and disease diagnosis. We design task-specific instructions to prompt the model to recognize and address different objectives. Considering the competitive performance of the open-source model InternVL 1.2 (Chen et al., 2024b), we utilize it as our general-domain foundation to begin training, which is composed of InternViT-6B as the vision encoder, and Nous-Hermes-2-Yi-34B as the language model. Our training process is divided into two steps: alignment training and instruction tuning. During the alignment training phase, we freeze the vision encoder and language model, only fine-tuning the alignment module with medical image captioning datasets. In the instruction tuning stage, we apply both public datasets and our Region-Centric datasets, MedRegInstruct, to optimize the language model, while keeping the other components unchanged. The language model loss is applied as the loss function. More training details can be found in Appendix B.

**Regional CoT**. After training on Region-Centric tasks, the model has been equipped with region identification and detection abilities, which have the potential to further enhance model performance on other general tasks. As proved in Zhang et al. (2023b); Yang et al. (2023), textualizing additional guidance such as spatial coordinates into the prompt would unleash the language model's ability to process multimodal information. Following these insights, we incorporate regional CoT into the generation pipeline to better leverage these regional skills, as illustrated in Figure 5. The model is required to initially detect critical regions in the input image, followed by generation prompted by the detected regions. This approach could encourage the model to attend more to internal structures within the medical scan when answering patient consultations or diagnosing diseases.

Table 1: **Performance comparison on general medical tasks and region-centric tasks.** 'N/A' means the scores are not reported. '-' indicates the model cannot generate valid outputs. '✗' denotes the model cannot tackle corresponding tasks. '*' means that the model is fine-tuned on the dataset.

| Task | Metrics | Med-Flamingo | LLaVA-Med | RadFM | MedDr | BiomedGPT | InternVL | MedRegA |
|---|---|---|---|---|---|---|---|---|
| *Visual Question Answering* | | | | | | | | |
| **English Visual Question Answering** | BLEU-1 | 21.16 | 54.59* | 51.19 | 65.72 | N/A | 63.51 | **67.65** |
| | F1 Score | 22.28 | 54.81* | 51.36 | 66.87 | N/A | 65.05 | **68.33** |
| | CloseAccuracy | 41.91 | 43.73* | 61.01 | 84.03 | **86.40*** | 64.99 | 83.25 |
| | OpenRecall | 12.03 | **63.51*** | 42.63 | 49.81 | 57.73* | 44.17 | 53.35 |
| | BertScore | 53.20 | 75.73* | 73.29 | 77.80 | N/A | 35.47 | **84.67** |
| **Chinese Visual Question Answering** | BLEU-1 | 10.28 | 22.17 | 11.30 | 37.76 | | 39.73 | **60.89** |
| | F1 Score | 11.44 | 22.32 | 11.63 | 39.86 | | 40.95 | **62.66** |
| | Accuracy | 13.46 | 16.50 | 12.32 | 35.53 | N/A | 38.02 | **58.64** |
| | Recall | 16.85 | 23.23 | 14.43 | 38.77 | | 40.20 | **62.01** |
| | BertScore | 59.10 | 28.24 | 65.75 | 87.55 | | 87.58 | **91.63** |
| *Report Generation* | | | | | | | | |
| **English Report Generation** | BLEU-1 | 21.50 | 17.47 | 29.06 | 34.49 | N/A | 13.12 | **40.46** |
| | BLEU-4 | 2.63 | 0.28 | 4.30 | 10.02 | N/A | 0.53 | **12.60** |
| | METEOR | 17.30 | 12.68 | 19.38 | 27.70 | 13.55* | 14.26 | **31.94** |
| | ROUGE-L | 13.79 | 8.54 | 13.74 | 24.61 | 26.45* | 10.56 | **27.57** |
| | BertScore | 49.81 | 44.34 | 53.43 | 55.12 | N/A | 49.26 | **62.54** |
| **Chinese Report Generation** | BLEU-1 | 3.59 | 4.91 | 3.66 | 11.99 | | 10.71 | **40.76** |
| | BLEU-4 | - | - | 0.02 | 0.32 | | 1.85 | **18.74** |
| | METEOR | 2.59 | 3.60 | 4.05 | 10.71 | N/A | 9.68 | **34.03** |
| | ROUGE-L | 2.98 | 3.58 | 4.46 | 8.44 | | 12.90 | **22.98** |
| | BertScore | 58.16 | 57.22 | 62.54 | 61.39 | | 61.14 | **71.11** |
| *Medical Image Classification* | | | | | | | | |
| **Single-Label Classification** | F1 Score | 16.99 | 22.84 | 18.29 | 32.65 | N/A | 21.13 | **47.97** |
| **Multi-Label Classification** | F1 Score | 2.53 | 5.27 | 8.32 | 9.38 | | 5.16 | **13.32** |
| *Region-Centric Tasks* | | | | | | | | |
| **Region-to-Text Identification** | BLEU-1 | 0.13 | 0.43 | 0.35 | 0.75 | | 0.13 | **69.72** |
| | F1 | 0.72 | 1.15 | 0.80 | 1.27 | | 0.21 | **70.43** |
| | Recall | 4.90 | 10.69 | 4.75 | 3.52 | N/A | 0.52 | **71.05** |
| | Accuracy | 0.01 | 1.74 | 1.88 | 1.36 | | - | **66.24** |
| | BertScore | 24.87 | 34.53 | 37.07 | 50.28 | | 49.85 | **87.13** |
| **Text-to-Region Detection** | Object F1 | ✗ | ✗ | ✗ | ✗ | ✗ | 56.60 | **77.93** |
| | Region F1 | ✗ | ✗ | ✗ | ✗ | ✗ | 6.70 | **38.24** |
| | Alignment F1 | ✗ | ✗ | ✗ | ✗ | ✗ | 5.45 | **36.53** |
| | IoU | ✗ | ✗ | ✗ | ✗ | ✗ | 12.28 | **23.43** |
| **Grounded Report Generation** | Report BLEU-1 | 19.41 | 17.36 | 22.32 | 27.43 | N/A | 19.46 | **33.18** |
| | Region Acc | ✗ | ✗ | ✗ | ✗ | ✗ | 0.57 | **76.59** |
| | Alignment Acc | ✗ | ✗ | ✗ | ✗ | ✗ | 0.12 | **62.29** |
| | IoU | ✗ | ✗ | ✗ | ✗ | ✗ | 0.38 | **52.07** |

# 5 EXPERIMENT

## 5.1 PERFORMANCE ON GENERAL MEDICAL TASKS

To evaluate the ability of MedRegA on general medical tasks, namely visual question answering, report generation and classification, we comprehensively compare our model with the base model InternVL (Chen et al., 2024c) and open source medical MLLMs in general domain, including Med-Flamingo (Moor et al., 2023), LLaVA-Med (Li et al., 2024), RadFM (Wu et al., 2023), MedDr (He et al., 2024), and BiomedGPT (Zhang et al., 2024a) by reproducing their released checkpoints with official prompting instructions. An overall performance comparison is shown in Table 1 with averaged results on each task. It should be notified that the existing baselines cannot manage to locate highlighted areas within the medical scan and provide valid regional outputs, exposing a huge limitation in differentiating body structures and detecting fine-grained abnormalities.

**Performance on Visual Question Answering.** In Visual Quesion Answering task, the model needs to answer questions involving the modality, visible structures, and possible diseases of the given medical scan. We perform comparison on several medical VQA benchmarks, such as English and Chinese versions of SLAKE (Liu et al., 2021), VQA-RAD (Lau et al., 2018), and PathVQA (He et al., 2020). The averaged results on these datasets revealing that MedRegA outperforms all the baselines in the overall metrics by approximately 2% to 40% in English and over 10% in Chinese.

**Performance on Medical Report Generation.** Medical Report Generation task requires the model to generate a detailed report based on the provided medical scan. For English report generation, we evaluate our model on the report generation task for chest X-ray datasets MIMIC-CXR (Johnson et al., 2019) and IU-Xray (Demner-Fushman et al., 2016). Moreover, to evaluate the bilingual ability of our model in report generation, we collect a Chinese report generation test dataset covering brain, chest, spine, abdomen and pelvis from the hospital. Table 1 shows that MedRegA excels all other

Table 2: **Performance on the region-to-text identification task.**

| Subtask | Metrics | Med-Flamingo | LLaVA-Med | RadFM | MedDr | InternVL | MedRegA |
|---------|---------|--------------|-----------|-------|-------|----------|---------|
| **Structure Identification** | BLEU-1 | 0.24 | 0.35 | 0.22 | 0.93 | 0.13 | **78.34** |
| | F1 | 1.33 | 0.94 | 0.44 | 1.64 | 0.21 | **78.95** |
| | Recall | 9.4 | 7.33 | 2.00 | 4.66 | 0.52 | **79.02** |
| | Accuracy | - | 1.3 | 2.62 | 2.71 | - | **73.06** |
| | BertScore | 26.58 | 34.88 | 38.28 | 51.18 | 51.78 | **91.63** |
| **Lesion Identification** | BLEU-1 | 0.01 | 0.51 | 0.47 | 0.57 | 0.13 | **61.09** |
| | F1 | 0.10 | 1.35 | 1.15 | 0.89 | 0.20 | **61.9** |
| | Recall | 0.39 | 14.05 | 7.49 | 2.37 | 0.51 | **63.07** |
| | Accuracy | 0.02 | 2.18 | 1.14 | - | - | **59.42** |
| | BertScore | 23.16 | 34.18 | 35.86 | 49.38 | 47.92 | **82.62** |

Precision = #Detected / #All , Recall = #Detected / #Predicted

Figure 6: **Definition of Region-Level, Object-Level, and Object-Region Alignment evaluation.** Boxes labeled with roman numerals denote groundtruths and those with capital letters denote predictions.

baselines in generating reports with both English and Chinese, achieving 5.97% and 28.77% higher averaged BLEU-1 score compared with MedDr, respectively.

**Performance on Medical Image Classification.** In Medical Image Classification task, the model is expected to either classify the medical structure shown in the image or diagnose the specific diseases indicated by the scan. We conduct experiments on both single-label classification and multi-label classification with a wide range of datasets across radiology, ultrasound, ophthalmology, and dermatology. We report the averaged F1 score in Table 1. For single-label classification, MedRegA outperforms existing models by a large margin from 15.32% to 30.98%. Multi-label classification appears more challenging for MLLMs due to the difficulties in decoupling subtle symptoms and relating them to corresponding diagnoses. To enhance the model's focus on disease lesions, we employ Regional CoT into multi-label classification, and present the results in Figure 2 (b). Specifically, with Regional CoT, the model achieved an F1 score of 61.75% on VinDr-SpineXR dataset, surpassing MedDr and MedRegA without Regional CoT by and 34.95% and 31.59%, respecitvely. The improved results indicate that incorporating regional information assists medical MLLMs in establishing explicit relationships between local regions and each class label, rather than considering the entire global image with all labels (as illustrated in Figure 5), which further enhances image perception and handles multi-label classification.

## 5.2 Region-Aligned Evaluation on Region-Centric Tasks

To evaluate the regional ability of MedRegA, we implement our model on the proposed Region-Centric Tasks. To quantitatively evaluate the regional perception and comprehension capabilities of medical MLLMs, we introduce a Region-Aligned evaluation framework to measure the model performance on these tasks. In this section, we first present the assessment metrics for each task and demonstrate the experimental results. Due to the inability of existing open-souce medical MLLMs to generate precise coordinates, we apply an MLLM with an impressive regional ability on natural images, InternVL (Chen et al., 2024b), as the compared baseline.

### 5.2.1 Evaluation on Region-to-Text Identification

**Settings**. In the Region-to-Text Identification task, the model is required to identify the name of the specified region based on the corresponding bounding box. Since the output for this task is in the form of pure texts, we adopt Natural Language Generation (NLG) metrics for performance measurement, including BLEU-1, F1 score, Recall, Accuracy and BertScore.

Table 3: **Performance on the text-to-region detection task.** 'N/A' means that the task does not correspond to the evaluation dimension.

| Subtask | Model | Object-Level Metrics | | | Region-Level Metrics | | | Object-Region Alignment Metrics | | | IoU |
|---|---|---|---|---|---|---|---|---|---|---|---|
| **Single-object Single-region** | InternVL | N/A | | | **Accuracy** | | | N/A | | | 16.61 |
| | | | | | 7.24 | | | | | | |
| | MedRegA | | | | 45.11 | | | | | | **42.70** |
| **Single-object Single-region** | InternVL | N/A | | | Precision | Recall | F1 Score | N/A | | | 13.99 |
| | | | | | 5.40 | 5.43 | 5.41 | | | | |
| | MedRegA | | | | 31.15 | 25.48 | 24.78 | | | | **31.63** |
| **Multi-object Single-region** | InternVL | Precision | Recall | F1 | Precision | Recall | F1 Score | Precision | Recall | F1 Score | 14.48 |
| | | 65.02 | 71.67 | 67.13 | 7.86 | 10.56 | 8.69 | 6.28 | 7.18 | 6.58 | |
| | MedRegA | 87.80 | 87.85 | 87.82 | 48.11 | 48.13 | 48.12 | 44.91 | 44.93 | 44.92 | **46.12** |
| **Multi-object Multi-region** | InternVL | Precision | Recall | F1 | Precision | Recall | F1 Score | Precision | Recall | F1 Score | 4.05 |
| | | 43.42 | 53.40 | 46.06 | 4.90 | 7.49 | 5.44 | 4.08 | 5.04 | 4.32 | |
| | MedRegA | 62.90 | 81.82 | 68.03 | 26.15 | 27.04 | 34.94 | 29.17 | 33.51 | 28.14 | **15.23** |

Table 4: **Performance on Chest X-ray Grounded Report Generation.** '✗' denotes the model cannot tackle corresponding tasks.

| Model | NLG Metrics | | | CE Metrics | | | Region-Aligned Metrics | | | IoU |
|---|---|---|---|---|---|---|---|---|---|---|
| | BLEU-1 | ROUGE-L | BertScore | CheXBert | RadGraph | RadCliq↓ | Object | Region | Alignment | |
| **MedDr** | 27.43 | 18.23 | 53.00 | 25.37 | 14.76 | 1.14 | ✗ | ✗ | ✗ | ✗ |
| **InternVL** | 19.46 | 12.20 | 16.07 | 26.75 | 9.33 | 1.42 | 0.23 | 0.57 | 0.12 | 0.38 |
| **MedRegA** | **33.18** | **21.64** | **55.37** | **39.00** | **25.23** | **0.96** | **62.65** | **76.59** | **62.29** | **52.07** |

| Metrics | MedDr | InternVL | MedRegA |
|---|---|---|---|
| BLEU-1 | 11.79 | 5.76 | **20.04** |
| ROUGE-L | 8.38 | 4.52 | **14.01** |
| BertScore | 66.59 | 63.22 | **74.11** |
| Region Acc | ✗ | - | **25.80** |
| IoU | ✗ | 2.06 | **35.63** |

Table 5: **Performance on Lesion Grounded Report Generation.** '✗' denotes the model cannot tackle corresponding tasks. '-' indicates the model cannot generate valid outputs.

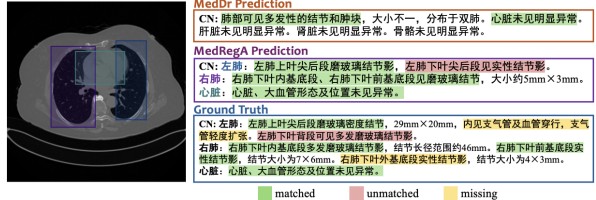

Figure 7: **Examples on Organ Grounded Report Generation.** The predicted bounding boxes are labeled with the same color as the detected organ in generated texts.

**Results.** We categorize the task into (1) structure identification for identifying anatomies such as structures within the brain, heart, lung, abdomen or spine, and (2) lesion identification focusing on abnormalities like tumors and cancers. Results of these two subtasks are reported in Table 2. It can be observed that MedRegA is able to accurately recognize the regions delineated by bounding boxes and associate them with the corresponding areas with 39.85% higher BertScore than InternVL, while previous methods fail to comprehend region encodings in medical scans. Notably, the model demonstrates a stronger capability of identifying body structures compared to lesions, possibly because anatomies tend to be larger, whereas lesions are subtler and exhibit more variation in shape.

### 5.2.2 EVALUATION ON TEXT-TO-REGION DETECTION

**Settings.** In the Text-to-Region Detection task, the model should detect the region corresponding to specific organs or anomalies. The input textual instruction contains the name of single or multiple objects, and the model is expected to output bounding boxes for all the relevant regions. We classify the task into four categories according to the number of detected objects and the number of regions per object: single-object single-region, single-object multi-region, multi-object single-region, and multi-object multi-region, as illustrated in Appendix A.1.1. We evaluate the performance across three dimensions: (1) *Object-Level*: assessing the correctly identified objects; (2) *Region-Level*: assessing accurately detected regions; (3) *Object-Region Alignment*: assessing whether the detected boxes are correctly aligned with the corresponding object. For single-object detection, all the output boxes are aligned with the given object, where evaluating the detection performance from the region-level is sufficient. The region is correctly detected if its Intersection over Union (IoU) score exceeds the threshold of 0.5. For multi-object detection, the alignment of text and box must also be considered. Figure 6 illustrates the three-dimensional evaluation in the multi-object setting.

**Results.** Table 3 demonstrate the model performance for Text-to-Region Detection. When detecting objects corresponding to a single region, our model achieves relatively high scores, with detection accuracy approaching 50%. However, when multiple regions are associated with the given objects, the detection task becomes more complex for MLLMs. In such cases, recall is higher than precision, inferring that the model suffers from difficulty in detecting all the required regions thoroughly and tends to conservatively generate fewer bounding boxes.

### 5.2.3 PERFORMANCE ON GROUNDED REPORT GENERATION.

**Chest X-ray Grounded Report Generation.** For the grounded reports constructed from MIMIC-CXR, we follow the same training-test split as the report generation task and evaluate the model with those accompanied by organ bounding boxes, consisting of 3,022 samples in total. The model performance on region detection is assessed as the multi-region single-object sub-task defined in §5.2.2. Furthermore, we integrate NLG and CE metrics to estimate the quality of the descriptions for each region. As shown in Table 4, our model can detect the chest structures and generate descriptions simultaneously with impressive scores on both report generation and region detection.

**Lesion Grounded Report Generation.** To evaluate our model performance on generating grounded reports that concentrate on the lesion area, we sample 438 pairs of tumor coordinates and corresponding Chinese descriptions annotated by doctors from our collected data. The evaluation metrics can be referred to as the single-region single-object sub-task defined in §5.2.2 since each medical scan is associated with a single area of abnormality. NLG metrics are applied to evaluate the descriptions. Table 5 demonstrates that our model outperforms established baselines, especially with 33.57% higher IoU than InternVL.

**Organ Grounded Report Generation.** As illustrated in §3.2, we enlarge the collected Chinese image-report data with automatically-annotated organs to enrich the report-grounded data. Through self-training with these data, MedRegA has the capability to yield grounded reports with detailed descriptions on organs, as the examples in Figure 5. Without training on region-centric tasks, MedDr can only generate a broad description for each organ and fail to relate fine-grained descriptions with organ coordinates. In contrast, MedRegA managed to ground generated sentences with corresponding organs, and generated more detailed description on the internal condition of the organ. Though there are subtle deviation in the nodule position, MedRegA still provided a thorough insight on the medical scan, exhibiting a remarkable ability to leverage regional information.

## 6 CONCLUSION

In this paper, we present an interpretable bilingual generalist medical AI system with an additional intention to enhance the ability of medical MLLMs to investigate critical regions within the given medical scan. From data aspect, we first formulate Region-Centric tasks tailored to region identification, detection, and fine-grained grounded report generation for each recognized regions. Specifically based on these tasks, we construct a large-scale dataset, MedRegInstruct, with a semi-automatic labeling system. In terms of model, we propose a Region-Aware medical MLLM, MedRegA, by leveraging our constructed dataset and multimodal medical corpora across diverse tasks for training. Experiments demonstrate that MedRegA achieves promising results on both image and region-level tasks. We believe that region-centric capability is essential in medical MLLM development, since establishing relations between specific regions and generated texts would not only encourage the model to focus on critical areas, but also facilitate interpretability and clinical interactivity.

### ACKNOWLEDGEMENT

This work was supported by grants from the National Natural Science Foundation of China (NSFC) and the Research Grants Council (RGC) of Hong Kong under the Joint Research Scheme (JRS), Project No. N_HKUST654. It was also supported by the Research Grants Council of the Hong Kong Special Administrative Region, China (Project Reference Number: T45-401/22-N).

ETHICS STATEMENT

The in-house dataset used in our research consists of existing retrospective data collected from the hospital, and is merely intended to advance research in healthcare. All data have been anonymized. Personal identifiers such as names, addresses, and contact details have been removed to protect the privacy and confidentiality of individuals. The data is ensured to be analyzed and interpreted responsibly to avoid any potential negative impact on patients or individuals. We will only release the constructed data derived from public datasets. We understand that we are solely responsible for any legal violations with respect to our benchmark, and we accept all the risks associated with our research and any data we release.

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

APPENDIX OF "INTERPRETABLE BILINGUAL MULTIMODAL LARGE LANGUAGE MODEL FOR DIVERSE BIOMEDICAL TASKS"

# A  DATASET DETAILS

## A.1  MEDREGINSTRUCT DATASET

The motivation of creating the MedRegInstruct Dataset lies in the lack of region-centric abilities and interoperability in existing medical MLLMs. In other words, existing medical MLLMs are global-centric, designed to treat the entire image as a holistic representation and struggling to detect or reason about specific regions within a medical scan. As a result, current medical generalist models cannot manage to concentrate on a critical region within the given image, thus leading to inaccurate descriptions specifically in the lesion location. Besides, merely generating diagnostics or reports without fine-grained locations is less interpretable, making it hard for clinicians to trace the source of the generated text.

Thus, we first establish Region-Centric tasks, aimed at improving medical MLLMs' capability to encode regional information. Then, we develop a large-scale dataset, MedRegInstruct, covering all those tasks in order to train an interpretable generalist medical MLLM with region-centric abilities.

### A.1.1  REGION-CENTRIC TASK

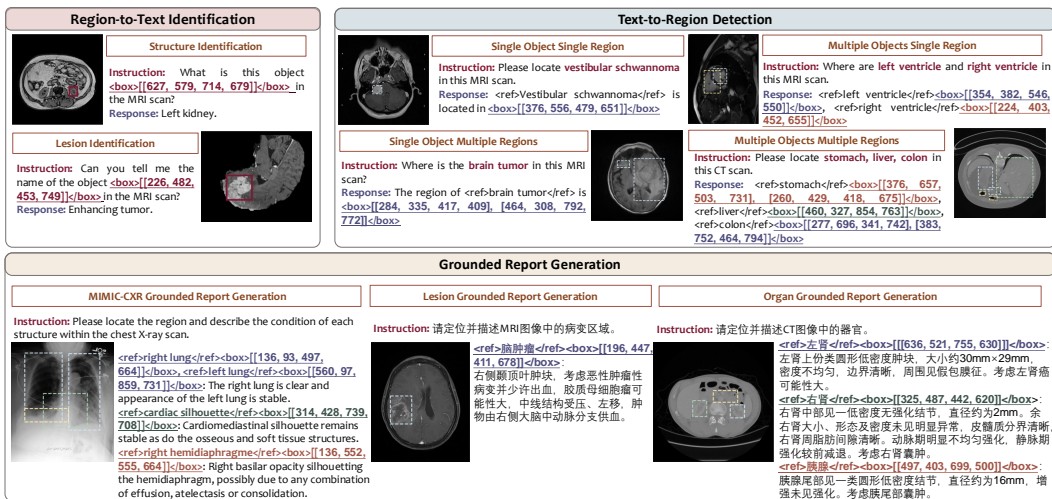

Figure 8: **Region-Centric Task Fomulation.** In the Region-to-Text Identification task, each question-answer pair contains one region to recognize. The detected region is composed of two types, body structure and disease lesion. In the Text-to-Region detection task, the model would be asked to simultaneously locate multiple regions for an object or multiple objects. Based on the number of detected objects and the number of regions per object, the detection task can be categorized into four subtasks, namely single-object single-region, single-object multi-region, multi-object single-region, and multi-object multi-region. In the Groubded Report Generation task, each region should be matched with a detailed description, discussing the condition of the located organ or lesion.

As illustrated in 8, we propose three types of Region-Centric tasks, (1) Region-to-Text identification: recognizing the name or condition of a given region; (2) Text-to-Region Detection: locating the position of structures mentioned in the instruction; (3) Grounded Report Generation: providing detailed reports for all highlighted anatomies along with their corresponding regions in the medical scan.

Based on the specific tasks in mind, we construct the MedRegInstruct dataset, consisting of the Region-Text Dataset for Region-to-Text Identification task and Text-to-Region detection task, and the Report-Grounded Dataset for Grounded Report Generation task.

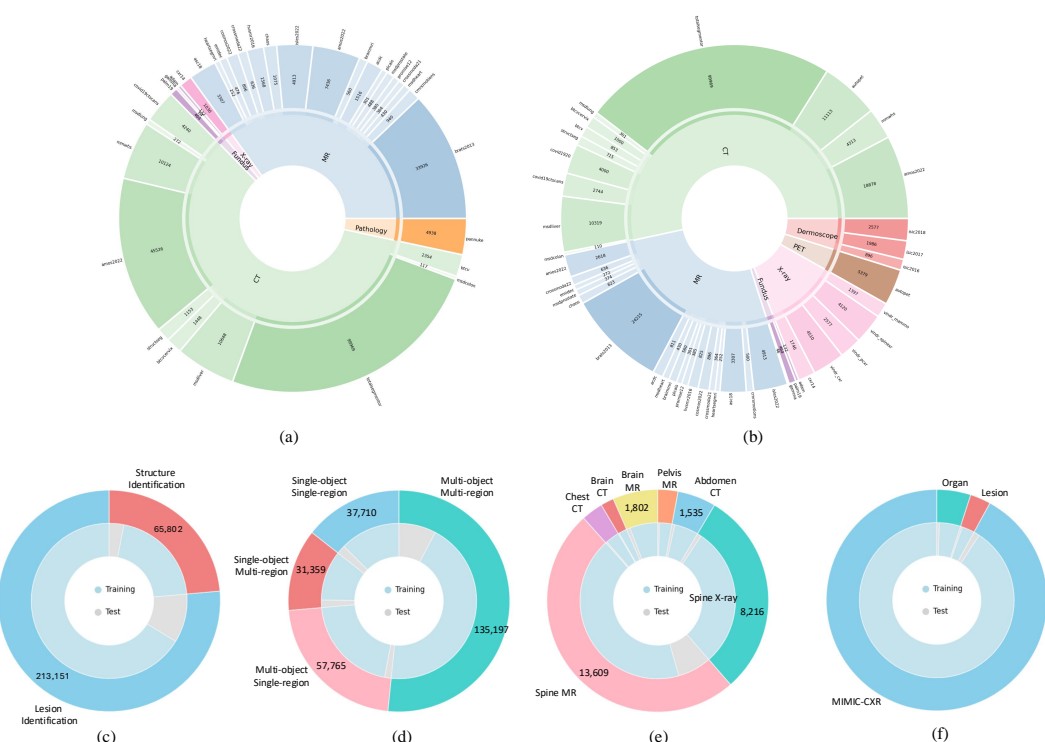

Figure 9: **MedRegInstruct dataset overview.** (a) and (b) demonstrate the distribution over modalities and datasets for Region-to-Text and Text-to-Region data, respectively. (c) and (d) presents the subtask distribution and training-test splits for Region-Text Dataset. (e) shows the body structure distribution of the collected report data. (f) exhibits the statistics of Report-Grounded dataset.

### A.1.2 REGION-TEXT DATASET

The Region-Text dataset is sourced from SA-Med2D-20M (Ye et al., 2023), from which approximately 285K images are filtered to construct into image-text-region triplets. To formulate the data into region-to-text identification and text-to-region detection task, we first employed GPT-4 to predefine 50 templates for each. Template examples are shown in Figure 10.

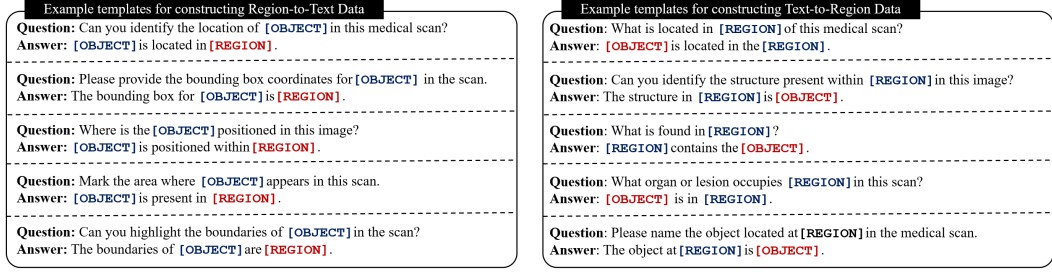

Figure 10: **Template examples to formulate triplets of (image, text, region) into region-to-text and text-to-region tasks.**

In statistics, the region-text dataset is composed of 278,923 samples for region-to-text task and 262,031 samples for text-to-region task, respectively.

### A.1.3 Report-Grounded Dataset

The Report-Grounded dataset sources from MIMIC-CXR and a clinical in-house dataset collected from the hospital, containing approximately 240K samples in total. Examples for the grounded reports can be found in Figure 8.

**In-house dataset for Chinese report generation**. Here we will provide a detailed overview of the collected clinical data. Our in-house clinical data containing approximately 25K X-ray, MRI and CT scans from 15K patients, each with a piece of report written in Chinese. The in-house data covers multiple regions, including the brain, chest, abdomen, spine, and pelvis, as shown in Figure 9. Also, various lesion types are considered, such as brain tumor, ischemic stroke, cerebral hemorrhage, nasopharyngeal carcinoma, vestibular schwannoma, lung cancer, pul- monary embolism, liver cancer, colon cancer, kidney cancer, pancreas cancer, prostate cancer, lum- bar degenerative disease, disc space narrowing, spondylolisthesis, etc. For scans in 3D volume, we select the central 2D slice that provides the most representative views of the anatomical structures. In total, our in-house dataset is composed of 27,357 image-report pairs. Examples are shown in Figure 11.

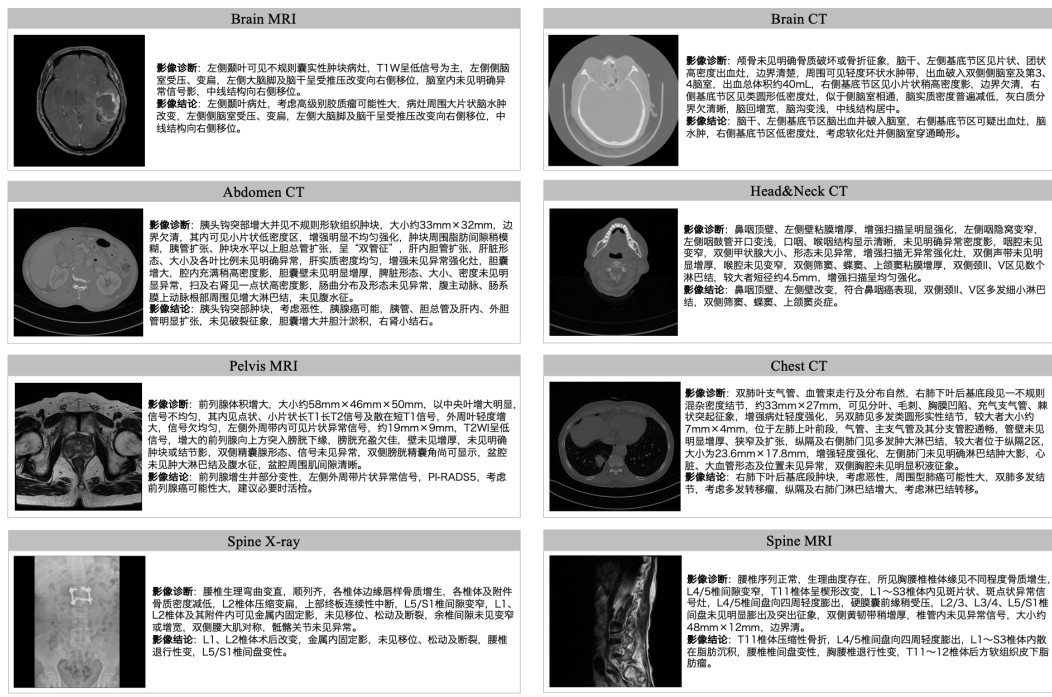

Figure 11: **Data examples of the Chinese image-report pairs in our collected dataset.**

**Grounded report construction from MIMIC-CXR dataset.** We apply Chest-ImaGenome (Wu et al., 2021), which includes bounding box annotations for 29 structures in the frontal chest x-ray images from MIMIC-CXR dataset. From these atanomies, we select 12 standardized ones in chest to create the report-grounded dataset for chest x-ray examination, including left lung, right lung, mediastinum, cardiac silhouette, left hilar structures, right hilar structures, left clavicle, right clavicle, left hemidiaphragm, right hemidiaphragm, right atrium, and abdomen. We filtered the chext x-ray scans paired with annotation boxes, and obtained 225,610 data samples in total.

**Grounded report construction In-house dataset**. The constructed report-grounded dataset from our in-house dataset contains two parts: (1) Lesion Grounded Report and (2) Organ Grounded Report. In our in-house dataset, certain scans are manually labeled with the lesion area by experts. This lesions include 12 types in total, namely, brain tumor, ischemic stroke, cerebral hemorrhage, nasopharyngeal carcinoma, vestibular schwannoma, lung cancer, pulmonary embolism, liver cancer, colon cancer, kidney cancer, pancreas cancer, prostate cancer. For these scans, we match the annotated abnormal region coordinates with specific descriptions for the lesion part, acquiring 6,881 grounded reports specifically for abnormalities. For the rest cases, we follow the semi-automatical

pipeline illustrated in §3.2 to construct organ grounded reports. The organ grounded report data covers body parts including heart, lung, liver, colon, kidney, spleen, and pancreas. We selected 5 slices from each original 3D scan to obtain more diverse views, and obtain 10,907 data items for organ grounded reports.

## A.2 TRAINING DATASETS ON TRADITIONAL MEDICAL TASKS

### A.2.1 IMAGE CAPTIONING

**PMC-OA** (Lin et al., 2023) is a biomedical dataset with 1.6M image-caption pairs collected from PubMedCentral's OpenAccess subset which covers diverse modalities or diseases. We leveraged the dataset for alignment training.

**QUILT** (Ikezogwo et al., 2024) is a large-scale vision-language dataset consisting of 802,144 image and text pairs, which was curated from video frames and corresponding subtitles on YouTube. We leveraged the dataset for alignment training.

### A.2.2 VISUAL QUESTION ANSWERING

**SLAKE** (Liu et al., 2021) is a bilingual radiology VQA dataset consists of 642 radiology images and over 7000 diverse QA pairs annotated by experienced physicians. Following the official split, we used both English and Chinese versions for training, which contains 4919 and 4916 question-answer pairs repectively.

**VQA-RAD** (Lau et al., 2018) consists of 3.5K question-answering pairs on 314 radiology images, where clinicians asked naturally occurring questions about radiology images and provided reference answers. Following the official split, we use 3,064 question-answer pairs for training.

**PathVQA** (He et al., 2020) consists of 32,799 open-ended questions from 4,998 pathology images where each question is manually checked to ensure correctness. Every image is paired with several questions related to multiple aspects such as shape, color and location. Following the official split, we use 19,755 question-answer pairs for training.

**PMC-VQA** (Zhang et al., 2023a) consists of 1.6 million question-answer pairs, which is a large-scale medical visual question-answering dataset generated from PMC-OA. We combined two versions of PMC-VQA and use 329,551 question-answer pairs for training.

**MedPix** is collected from MedPix website[1], which is a free open-access online database for medical usage. RadFM (Wu et al., 2023) separate the dataset into MPx-single and MPx-multi. We apply the MPx-single part and use 92,282 question-answer pairs for training.

### A.2.3 REPORT GENERATION

**MIMIC-CXR** (Johnson et al., 2019) is a large-scale chest image-report dataset that consists of 371,920 chest X-rays associated with 227,943 reports from 65,079 patients. Following RadFM (Wu et al., 2023), we use 354,569 cases for training.

**IU-Xray** (Demner-Fushman et al., 2016) consists of 7,470 images and 3,955 reports collected from the Indiana Network. Following R2Gen (Chen et al., 2020), we use 4,720 cases for training.

**In-house Dataset**. We split the collected reports at the patient level, ensuring that the training and test sets do not contain overlapping patients. we use approximately 90% of all the data for training, which contains 24,608 image-report pairs.

### A.2.4 MEDICAL IMAGE CLASSIFICATION

**VinDr-CXR** (Nguyen et al., 2022) consists of 18,000 images that were manually annotated by a total of 17 experienced radiologists with 22 local labels of rectangles surrounding abnormalities and 6 global labels of suspected diseases. The training set contains 15,000 scans, and 3 radiologists independently label each image. Following the official split, we use 45,000 samples for training.

---

[1]https://medpix.nlm.nih.gov

**VinDr-PCXR** (Pham et al., 2022) is a pediatric CXR dataset of 9,125 studies that were retrospectively collected from a major pediatric hospital in Vietnam between 2020-2021. Each scan was manually annotated by an experienced radiologist for the presence of 36 critical findings and 15 diseases. Following the official split, we use 4,585 samples for training.

**VinDr-SpineXR** (Nguyen et al., 2021) is a large-scale annotated medical image dataset for spinal lesion detection and classification from radiographs. The dataset contains 10,466 spine X-ray images from 5,000 studies, each of which is manually annotated with 13 types of abnormalities by an experienced radiologist with bounding boxes around abnormal findings. Following RadFM (Wu et al., 2023), we use 8,389 samples for training.

**VinDr-Mammo** (Nguyen et al., 2023) is a large-scale full-field digital mammography dataset of 5,000 four-view exams. Following the official split, we use 16,391 samples for training.

**CheXpert** (Irvin et al., 2019) is a large public dataset for chest radiograph interpretation, which retrospectively collected the chest from Stanford Hospital, performed between October 2002 and July 2017. The dataset contains 224,316 chest radiographs of 65,240 patients. Following the official split, we use 223,414 samples for training.

**MURA** (Rajpurkar et al., 2017) is a large-scale dataset of musculoskeletal radiographs containing 40,561 images from 14,863 studies, where each study is manually labeled by radiologists as either normal or abnormal. Following the official split, we use 36,808 samples for training.

**ISIC2018** (Codella et al., 2019) is a skin lesion dataset acquired with 7 dermatoscope types. Following the official split, we use 10,015 samples for training.

**ISIC2019** (Combalia et al., 2019) is a skin lesion dataset labeled with 8 different categories. Following the official split, we use 25,331 samples for trianing.

**PAD-UFES** (Pacheco et al., 2020) is a skin lesion dataset composed of clinical images collected from smartphone devices and a set of patient clinical data containing up to 22 features. The dataset consists of 1,373 patients, 1,641 skin lesions, and 2,298 images for six different diagnostics. We randomly sample 80% of the dataset for training, which includes 1,838 samples.

**Kather colon dataset** (Kather et al., 2019) is a dataset of 100,000 non-overlapping image patches from hematoxylin & eosin (H&E) stained histological images of human colorectal cancer (CRC) and normal tissue, covering 9 tissue classes in total.

**BRSET** (Nakayama et al., 2023) is a multi-labeled ophthalmological dataset onsisting of 16,266 images from 8,524 Brazilian patients. Multi-labels are included alongside color fundus retinal photos. We randomly sample 80% of the dataset for training, containing 13,012 samples.

**ODIR-5K** (Li et al., 2021) is a structured ophthalmic database of 5,000 patients with age, color fundus photographs from left and right eyes and doctors' diagnostic keywords from doctors. Following the official split, we use 6,392 samples for training.

**OCT2017** (Kermany et al., 2018) includes 83,484 OCT images of 4,686 patients, consisting of 4 categories, normal, drusen, choroidal neoVascularisation (CNV), and Diabetic Macular Edema (DME). Following the official split, we use 82,484 samples for training.

**Butterfly Network ultrasound dataset** (ButterflyNetworkInc., 2018) is a large dataset containing 9 different classes of ultrasound images acquired with the Butterfly IQ on 31 individuals. Following Chen et al. (2021), 34,325 images are applied for training.

**BUSI** (Al-Dhabyani et al., 2020) includes breast ultrasound images among women between 25 and 75 years old. The number of patients is 600 females, patients. The dataset consists of 780 images that are categorized into three classes, namely, standard, benign, and malignant. We randomly sample 80% of the dataset for training, which includes 630 images.

## A.3 TRAINING DATASETS ON REGION-CENTRIC TASKS (MEDREGINSTRUCT)

### A.3.1 REGION-TEXT TASKS

**SA-Med2D-20M** (Ye et al., 2023) is a large-scale segmentation dataset of 2D medical images built upon numerous public and private datasets. The dataset consists of 4.6 million 2D medical images

and 19.7 million corresponding masks, covering almost the whole body and showing significant diversity. We filter approximately 285K images from the original dataset, and construct 242,268 and 229,340 training samples for Region-to-Text Identification and Text-to-Region Detection respectively.

**VinDr Series Dataset** is a large-scale classification composed of VinDr-CXR, VinDr-PCXR, VinDr-SpineXR, VinDr-Mammo. The datasets provide radiologist's bounding-box annotation for abnormal areas. We follow the official split and apply the samples with bounding boxes.

**ISIC Challenge Dataset** contains lesion segmentation data where the original image is paired with manually annotated lesion boundaries. We follow the official split and convert the segmentation map into bounding boxes.

**PanNuke** (Gamper et al., 2019) is a semi-automatically generated nuclei instance segmentation dataset. We follow the official split and convert the segmentation map into bounding boxes to formulate region-text pairs.

### A.3.2 REPORT-GROUNDED TASKS

**Chest-ImaGenome** (Wu et al., 2021) applied a CXR bounding box detection pipeline to automatically label frontal chest x-ray images from MIMIC-CXR dataset with 29 annotations, from which we selected 12 standardized structures in the chest. Following the split of MIMIC-CXR in RadFM Wu et al. (2023), we filtered the chext x-ray scans paired with annotation boxes, and obtained 222,588 samples for training.

**In-house dataset**. The constructed report-grounded dataset from our in-house dataset contains two parts, Lesion Grounded Report and Organ Grounded Report. We applied 6,443 lesion grounded reports and 10,907 organ grounded reports for training.

### A.4 TEST DATATSETS

### A.4.1 VISUAL QUESTION ANSWERING

**SLAKE**. Following the official split, we use 1,061 and 1,033 quesion-answer pairs for test on the English and Chinese version, respectively.

**VQA-RAD**. Following the official split, we use 451 question-answer pairs for evaluation.

**PathVQA**. Following the official split, we use 6,761 quesion-answer pairs for evaluation.

**PMC-VQA**. Following the official split, the two versions of PMC-VQA test set contain 50,000 and 33,430 quesion-answer pairs, respectively. We test the model on both versions and report the averaged result.

### A.4.2 REPORT GENERATION

**MIMIC-CXR**. Following RadFM (Wu et al., 2023), we use 4,710 cases for test.

**IU-Xray**. Following R2Gen (Chen et al., 2020), we use 1,180 cases for training.

**In-house Dataset.** We split the collected reports at the patient level, ensuring that the training and test sets do not contain overlapping patients. we use approximately 10% of all the data for test, which includes 2,749 image-report pairs.

### A.4.3 MEDICAL IMAGE CLASSIFICATION

**MURA**. Following the official split, we use 3,193 X-ray images for test.

**PneumoniaMNIST**. Following the official split, we use 624 X-ray samples for test.

**OrganCMNIST**. Following the official split, we use 8,216 CT samples for test.

**OrganAMNIST**. Following the official split, we use 8,827 CT samples for test.

**ISIC2016**. Following the official split, we use 379 skin lesion images for test.

**ISIC2018**. Following the official split, we use 1,512 skin lesion images for test.

**PAD-UFES-20**. Since no official split is provided, we randomly split 20% of the data for evaluation, which contains 460 images.

**Kather Colon Dataset**. The original dataset is split into "NCT-CRC-HE-100K" and "CRC-VAL-HE-7K" subsets, which share no overlap with each other. Since the "NCT-CRC-HE-100K" subset is applied for training, we use "CRC-VAL-HE-7K" subset for test, which contains 7,180 pathology images.

**Messidor-2**. The Messidor-2 dataset is a collection of Diabetic Retinopathy (DR) examinations. We treat this task a binary classification to detect DR disease. Since no official split is provided, we randomly split 20% of the data for evaluation, which includes 378 fundus images.

**OCT2017**. Following the official split, we use 968 OCT scans for test.

**OCTMNIST**. Following the official split, we use 1,000 OCT scans for test.

**BUSI**. Since no official split is provided, we randomly split 20% of the data for test, which includes 150 ultrasound images.

**BreastMNIST**. Following the official split, we use 156 ultrasound scans for test.

**CheXpert**. Following the official split, we use 234 chest x-ray images for test.

**ChestMNIST**. Following the official split, we use 22,433 chest x-ray images for test.

**CXR14**. We follow the official split in Holste et al. (2023) and apply 21,081 chest x-ray images for test.

**VinDr-CXR**. Following the official split, we use 3,000 chest x-ray images for evaluation.

**VinDr-PCXR**. Following the official split, we use 1,397 chest x-ray images for evaluation.

**VinDr-SpineXR**. Following the official split, we use 2,077 spine x-ray images for evaluation.

**VinDr-Mammo**. Following the official split, we use 4,000 mammography images for evaluation.

**BRSET**. Since no official split is provided, we randomly split 20% of the data for test, which includes 3254 fundus images.

**RFMiD 1.0**. Following the official split, we use 640 fundus images for evaluation.

### A.4.4 REGION-CENTRIC TASKS

**Region-Text Tasks**. We split approximately 10% from the Region-Text dataset for evaluation, including 36,655 samples for Region-to-Text Identification and 32,691 samples for Text-to-Region Detection.

**Report-Grounded Tasks**. For chest x-ray grounded reports, we follow the split in RadFM (Wu et al., 2023) and filtered the chext x-ray scans paired with annotation boxes. Overall, 3,022 samples are acquired for evaluation. For lesion grounded reports and organ grounded reports, we include 438 and 1,264 data items in the test set, respectively.

## B TRAINING DETAILS

### B.1 IMPLEMENTATION DETAILS

We employ InternVL 1.2 (Chen et al., 2024b) as our general-domain foundation to begin training, which is composed of InternViT-6B as the vision encoder, and Nous-Hermes-2-Yi-34B as the language model. Our training process is divided into two steps: alignment training and instruction tuning. During the alignment training phase, we freeze the vision encoder and language model, only fine-tuning the alignment module with medical image captioning and report generation datasets, which contain about 2.4M data in total. In the instruction tuning stage, we apply both public datasets and our Region-Centric datasets, MedRegInstruct, to optimize the language model, while keeping the other components unchanged. The amount of instruction tuning data is approximately 2.2M. The

language model loss is applied as the loss function. We follow the official instruction for finetuning InternVL, and leverage LoRA with DeepSpeed ZeRO Stage 3 to optimize model parameters. The model is trained on 16 NVIDIA H800 GPUs for 1 epoch in the alignment stage and 2 epochs in the instruction tuning stage.

## B.2 TASK-SPECIFIC PROMPTS

We design task-specific instructions to prompt the model to recognize and address different objectives for visual question answering, report generation, image classification and region-centric tasks. The instruction prompt for each task can be found in Figure 12.

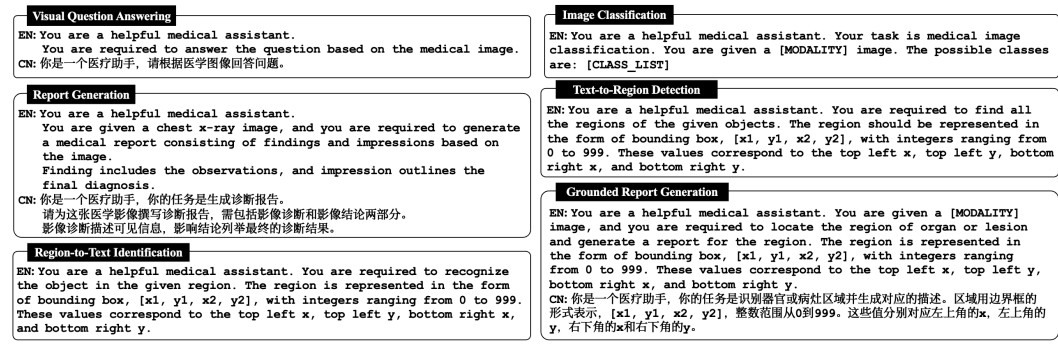

Figure 12: **Task-Specific Prompt Examples.**

## C EXPERIMENT DETAILS

### C.1 PERFORMANCE COMPARISON ON GENERAL MEDICAL VISION-LANGUAGE TASKS

#### C.1.1 VISUAL QUESTION ANSWERING

Table 6: **Performance comparison on Visual Question Answering task.** '*' indicates that the model is fine-tuned on the dataset. 'N/A' means the scores are not reported.

| Dataset | Modality | Metrics | Med-Flamingo | LLaVA-Med | RadFM | MedDr | BioMedGPT | Ours |
|---------|----------|---------|--------------|-----------|-------|-------|-----------|------|
| *Visual Question Answering* | | | | | | | | |
| SLAKE | X-ray, MRI, CT | BLEU-1 | 11.75 | 70.76* | 78.01 | 76.40 | N/A | **81.55** |
| | | F1 | 13.43 | 70.96* | 78.09 | 77.50 | N/A | **82.06** |
| | | BertScore | 40.22 | 84.89* | 87.33 | 78.71 | N/A | **91.76** |
| | | CloseAccuracy | 33.24 | 45.63* | 82.25 | 83.40 | 89.90* | 85.35 |
| | | OpenRecall | 21.06 | 83.80* | 76.23 | 74.20 | 84.30* | 80.45 |
| VQA-RAD | X-ray, MRI, CT | BLEU-1 | 27.4 | 44.6* | 50.7 | 59.34 | N/A | **61.89** |
| | | F1 | 28.22 | 44.77* | 51.04 | 60.99 | N/A | **62.24** |
| | | BertScore | 61.93 | 69.13* | 74.13 | 78.70 | N/A | **82.23** |
| | | CloseAccuracy | 44.62 | 27.39* | 60.56 | 78.49 | 81.3* | 75.30 |
| | | OpenRecall | 12.68 | 67.43* | 41.57 | 41.75 | 60.90* | 46.03 |
| PathVQA | Pathology | BLEU-1 | 24.31 | 48.39* | 24.84 | **61.40** | N/A | 59.52 |
| | | F1 | 25.18 | 48.7* | 24.96 | **62.10** | N/A | 60.69 |
| | | BertScore | 57.45 | 73.18* | 58.4 | 76.00 | N/A | **80.03** |
| | | CloseAccuracy | 47.86 | 58.18* | 40.22 | **90.20** | 88.00* | 89.12 |
| | | OpenRecall | 2.35 | **39.27*** | 10.08 | 33.50 | 28.00* | 33.56 |
| PMC-VQA | X-ray, MRI, CT | BLEU-1 | 30.32 | 39.19 | 48.71 | 64.15 | N/A | **67.19** |
| | | F1 | 33.27 | 41.98 | 49.47 | 64.83 | N/A | **67.88** |
| | | BertScore | 62.21 | 67.2 | 70.54 | 84.41 | N/A | **85.26** |
| | | CloseAccuracy | 35.88 | 63.82 | 45.76 | 66.18 | N/A | **81.18** |
| | | OpenRecall | 39.13 | 53.09 | 52.19 | 65.15 | N/A | **68.07** |

We implement our model on several medical VQA benchmarks for comparison, such as English and Chinese versions of SLAKE (Liu et al., 2021), VQA-RAD (Lau et al., 2018), PathVQA (He et al., 2020), and PMC-VQA (Zhang et al., 2023a). For PMC-VQA, we combined the data from both versions. We follow the official splits of all the datasets and apply both natural language generation (NLG) and classification metrics to evaluate the model outputs. Table 6 represents the results on

these datasets. Specifically, LLaVa-Med and BioMedGPT shows slightly strong performance on the SLAKE, VQA-RAD, and PathVQA datasets because they are fine-tuned on each of the dataset. RadFM is primarily trained on radiology data and struggle to understand pathology modalities, leading to poor performance on the PathVQA dataset. Compared with MedDr, our model achieves more competitive performance, indicating that incorporating regional information into the training data allows the MLLM to better perceive spatial knowledge of various organs and lesion areas in medical images. However, our model performs slightly less competitive on PathVQA, likely due to the scarcity of regional information for the pathology images. This highlights the urgency of enhancing region-centric capabilities across a wide range of modalities, which would also benefit traditional vision-language tasks.

### C.1.2 REPORT GENERATION

Table 7: **Performance comparison on English Report Generation task.** 'N/A' means the scores are not reported.

| Dataset | Modality | Metrics | | Med-Flamingo | LLaVA-Med | RadFM | MedDr | BioMedGPT | Ours |
|---------|----------|---------|---------|--------------|-----------|-------|-------|-----------|------|
| | | *Report Generation in English* | | | | | | | |
| MIMIC-CXR | X-ray | NLG | BLEU-1 | 21.67 | 20.62 | 30.92 | 31.27 | N/A | **35.18** |
| | | | BLEU-4 | 1.83 | 0.43 | 4.55 | 7.83 | 9.90 | **10.36** |
| | | | METEOR | 15.38 | 14.09 | 20.37 | 23.10 | 14.20 | **28.36** |
| | | | ROUGE-L | 13.1 | 10.36 | 14.83 | 20.92 | 24.40 | **24.70** |
| | | | BertScore | 46.46 | 43.86 | 53.14 | 53.05 | N/A | **58.79** |
| | | CE | CheXBert | 16.67 | 11.63 | 21.42 | 24.72 | N/A | **34.89** |
| | | | RadGraph | 8.92 | 4.85 | 13.15 | 20.28 | 22.50 | **22.67** |
| | | | RadCliq↓ | 1.41 | 1.6 | 1.2 | 1.15 | N/A | **0.85** |
| IU-Xray | X-ray | NLG | BLEU-1 | 21.33 | 14.31 | 27.2 | 37.70 | N/A | **45.73** |
| | | | BLEU-4 | 3.42 | 0.13 | 4.04 | 12.20 | N/A | **14.84** |
| | | | METEOR | 19.22 | 11.26 | 18.39 | 32.30 | 12.90 | **35.51** |
| | | | ROUGE-L | 14.47 | 6.72 | 12.65 | 28.30 | 28.50 | **30.43** |
| | | | BertScore | 53.15 | 44.81 | 53.71 | 57.19 | N/A | **66.28** |
| | | CE | CheXBert | 30.78 | 27.99 | 30.77 | 56.40 | N/A | **61.83** |
| | | | RadGraph | 13.63 | 2.85 | 11.69 | 33.10 | N/A | **38.00** |
| | | | RadCliq↓ | 1.1 | 1.41 | 1.11 | 1.01 | N/A | **0.27** |

**English Report Generation.** For English report generation, we evaluate our model on the report generation task for chest X-ray datasets MIMIC-CXR (Johnson et al., 2019) and IU-Xray (Demner-Fushman et al., 2016). We apply the training-test split of Wu et al. (2023) for MIMIC-CXR and Chen et al. (2020) for IU-Xray. Natural language generation (NLG) and Clinical Efficiency (CE) metrics are utilized for assessment. Specifically, DeBERTa is deployed to calculate BertScore. Among these models, Med-Flamingo and LLaVA-Med concentrate on question-answering skills but are less effective in producing longer sentences to thoroughly describe the observations. In comparison with RadFM and MedDr, we integrate the recognition and detection of chest X-ray structures into the training data, which inherently encourages the model to analyze fine-grained anatomical details in the medical scan, thereby accomplishing significantly better results. For instance, on the MIMIC-CXR dataset, our model outperforms 3.91% in BLEU-1 and 3.78 in ROUGE-L compared with MedDr. Besides, MedRegA suparsses BioMEdGPT by 1.93% in ROUGE-L on the IU-Xray dataset. Examples in Figure 13 demonstrate that our model can consistently provide detailed and accurate descriptions of the findings in chest x-ray, closely aligning with the ground truth annotations. These results highlight highlight the ability of MedRegA to produce coherent and clinically relevant findings.

**Evaluation on ReXrank Benchmark**. To provide a comparison of our methods with state-of-the-art methods for chest X-ray report generation in English, we evaluate our method with the official test dataset on ReXrank (Lab, 2024). We present the results in Table 9. Our model achieves results that are comparable to specialist models, such as RGRG (Tanida et al., 2023) and RaDialog (Pellegrini et al., 2023), which are specifically trained on report generation tasks for chest X-ray datasets. However, unlike these specialist models, our model is designed as a generalist framework capable of handling multiple tasks, such as VQA and diagnosis, and supporting more modalities. Within the generalist model category, our model ranks second. It accomplishes superior scores in certain metrics compared to MedVersa (Zhou et al., 2024), and outperformed other generalists including RadFM (Wu et al., 2023) and BioMedGPT (Zhang et al., 2024a). In comparison with MedVersa, our model offers broader versatility in report generation for different modalities of various organs,

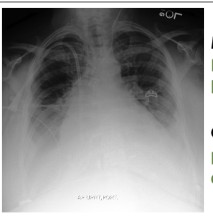

**MedRegA Prediction:** Findings: In comparison with the study of ___, there is little overall change. Monitoring and support devices remain in place. Continued enlargement of the cardiac silhouette with bilateral pleural effusions and compressive atelectasis at the bases.

**Ground Truth:** Findings: In comparison with the study of ___, the monitoring and support devices remain in place. Continued substantial enlargement of the cardiac silhouette with bilateral pleural effusions, compressive basilar atelectasis, and moderate pulmonary edema.

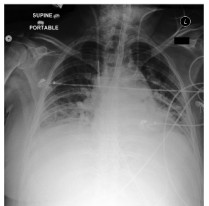

**MedRegA Prediction:** Findings: As compared to the previous radiograph, the monitoring and support devices are in unchanged position. The lung volumes have decreased. There is a mild-to-moderate pulmonary edema. The size of the cardiac silhouette is unchanged. No pleural effusions. No pneumothorax.

**Ground Truth:** Findings: As compared to the previous radiograph, there is no relevant change. The monitoring and support devices are constant. Low lung volumes, borderline size of the cardiac silhouette. Mild pulmonary edema. Moderate retrocardiac atelectasis. No evidence of pneumonia.

(a) MIMIC-CXR

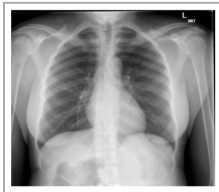

**MedRegA Prediction:** The heart is normal in size. The mediastinum is unremarkable. The aorta is tortuous. The lungs are hyperinflated. There is no pleural effusion, pneumothorax, or focal airspace disease. Mild degenerative changes are noted in the thoracic spine.

**Ground Truth:** The heart is normal in size. The mediastinum is stable. Atherosclerotic calcifications of the aorta identified. There is no focal consolidation, pleural effusion or pneumothorax. Degenerative changes of the thoracic spine are noted.

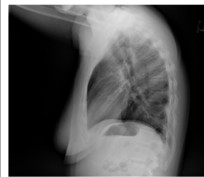

**MedRegA Prediction:** The heart is normal in size. The mediastinum is unremarkable. The lungs are clear. There is no pleural effusion or pneumothorax.
**Ground Truth:** The lungs are clear. There is no pleural effusion or pneumothorax. The heart and mediastinum are normal. The skeletal structures are normal

(b) IU-Xray

Figure 13: **Generated Report Examples on MIMIC-CXR and IU-Xray datasets.**

including X-ray, CT and MRI images of the brain, spine, abdomen, and pelvis, in addition to chest X-rays.

**Chinese Report Generation.** To evaluate the ability of our model in generating medical report in Chinese, we collect Chinese image-report pairs in real clinical scenarios to construct a test dataset covering brain, chest, spine, abdomen and pelvis from the hospital. Table 8 demonstrates the model performance comparison on Chinese report generation. Since MedDr reveals the ability to comprehend and generate Chinese language, we directly prompt it with Chinese texts to fetch the outputs. For other monolingual baselines, we obtain English reports and deploy an additional LLM to translate them into Chinese for evaluation. To quantify the capability of the models in Chinese report generation, we utilize NLG metrics to report the averaged result, where a Chinese tokenizer is also adopted to segment the generated sentences into word phrases. We apply Multilingual BERT to calculate the BertScore metric. MedRegA significantly outperforms other baselines in generating Chinese reports, revealing impressive bilingual ability. It can be inferred from the examples in Figure 14 that training on bilingual reports enables MedRegA to retain knowledge in both languages, yielding coherent and fluent medical reports in Chinese. For example, in multiple cases, MedRegA can successfully capture observations of lesions in a higher level of granularity, including size and locations.

**Evaluation on paired bilingual dataset.** Moreover, to evaluate the generalizability of our model across English and Chinese languages, We further apply GPT-4o to create paired bilingual datasets, including a English-to-Chinese dataset from IU-Xray and a Chinese-to-English dataset from our in-house data. Both datasets contain 100 samples respectively. Table 10 demonstrates that directly prompting MedRegA to generate report in a different language results in lower performance metrics. This can be attributed to the differences in writing style between the English and Chinese report

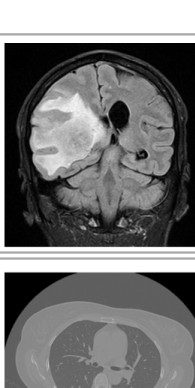

**MedRegA Prediction:**
影像诊断: 右侧颞叶、岛叶、枕叶见不规则肿块影, 大小约84mm×70mm, 病灶周围见大片状水肿带, 以右侧额叶区为著, 右侧侧脑室受压变窄, 中线结构向左侧移位, 余脑实质未见异常信号灶。
影像结论: 右侧颞叶、岛叶、枕叶肿块, 考虑恶性肿瘤性病变, 胶质母细胞瘤可能, 建议结合临床。

**Ground Truth:**
影像诊断: 右侧颞顶叶见不规则肿块影, 大小约为54.0mm×38.0mm×46.2mm(前后、左右、上下),
右侧侧脑室受压变窄, 对侧侧脑室扩大, 脑中线结构受压向左侧移位。
影像结论: 右侧颞顶叶占位性病变, 考虑恶性肿瘤性病变, 胶质母细胞瘤可能性大, 中线结构明显受压左移, 双侧额叶深部、放射冠、半卵圆中心多发腔隙性脑梗死及缺血灶, 右侧胚胎型大脑后动脉, 脑动脉硬化。

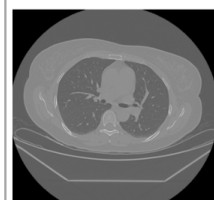

**MedRegA Prediction:**
影像诊断: 双肺叶支气管、血管束走行及分布自然, 右肺下叶后基底段见一部分实性结节影, 大小约18mm×17mm, 可见分叶、毛刺、胸膜凹陷, 增强扫描呈明显不均匀强化, 双肺散在数个实性结节影, 边界清, 较大者位于左肺下叶前基底段, 直径约4mm, 双肺散在数个囊状透亮影, 边界清, 较大者位于右肺上叶后段, 直径约10mm, 肺纹理稍粗, 双侧肺门结构清楚, 气管、主支气管及其分支管腔通畅, 管壁未见明显增厚、狭窄及扩张, 纵隔内结构清楚, 纵隔、双侧肺门未见明确肿大淋巴结影, 心脏、大血管形态及位置未见异常, 双侧胸腔未见积液征象, 所见肋骨、胸椎骨质未见明确异常。
影像结论: 右肺下叶后基底段实性结节, 考虑周围型肺癌可能, 双肺散在数个实性结节, 考虑增殖灶可能。

**Ground Truth:**
影像诊断: 双肺叶支气管、血管束走行及分布自然, 右肺下叶背段见一结节, 大小约22mm×20mm, 增强扫描呈轻度强化, 结节边界不清, 周围见毛刺, 邻近胸膜增厚粘连, 双肺多发结节, 较大者位于右肺下叶背段, 大小约7.1mm×6.3mm, 左肺上叶尖后段、上舌段、下叶背段及右肺上叶前段见多发磨玻璃结节, 较大者位于左肺上叶尖后段, 大小约7mm×5mm, 右肺中叶外侧段见少许片状磨玻璃密度增高影, 边界不清, 余肺内未见实质性或间质性病变, 双侧肺门结构清楚, 气管、主支气管及其分支管腔通畅, 管壁未见明显增厚, 纵隔、右肺门见数个小淋巴结, 较大者短径约7mm, 心脏、大血管形态及位置未见异常, 双侧胸腔未见明显积液征象。
影像结论: 右肺下叶背段结节, 符合肺癌, 双肺多发结节, 其中右肺下叶背段结节考虑转移瘤可能性大, 余结节不除外转移瘤可能, 纵隔、右肺门多发小淋巴结, 考虑转移结节, 左肺上叶尖后段、上舌段、下叶背段及右肺上叶前段多发磨玻璃结节, 考虑原位癌。

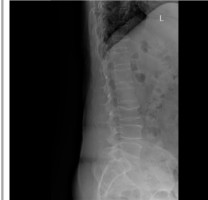

**MedRegA Prediction:**
影像诊断: 腰椎生理弯曲变直, 序列齐, 各椎体边缘不同程度增生变尖, 各椎体附件未见明确异常, l4/5椎间隙变窄, 余椎间隙未见明显变宽或变窄, 周围软组织未见异常, 动力位示腰椎过伸过屈活动受限, 未见椎体不稳或滑脱征象。
影像结论: 腰椎退行性变, l4/5椎间隙变窄, 拟椎间盘变性所致, 动力位示腰椎过伸过屈活动受限, 未见椎体不稳或滑脱征象。

**Ground Truth:**
影像诊断: 腰椎生理弯曲变直, 序列齐, 各椎体边缘不同程度增生变尖, 可见骨赘形成, t11及l3椎体压缩变扁, 余各椎体及附件未见明确异常, l4/5椎间隙变窄, 余各椎间隙未见变窄, 周围软组织未见异常密度影, 动力位示, 腰椎过伸过屈活动受限。
影像结论: 胸腰椎骨质疏松, t11及l3椎体压缩性骨折, 腰椎退行性变, l4/5椎间隙变窄, 考虑椎间盘变性, 动力位示, 腰椎过伸过屈活动受限。

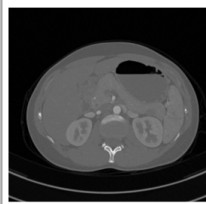

**MedRegA Prediction:**
影像诊断: 左肾上极见一不类圆形软组织致密影, 大小约40mm×35mm×32mm, 与左肾上份肾盂肾盏相连, 左肾上份肾盂、肾盏扩张, 左输尿管上段未见显影, 右肾及双侧肾上腺形态、大小、密度未见异常, 双侧肾盂、肾盏未见明显扩张积水, 膀胱形态、大小未见明显异常, 腔内未见明确异常密度影, 膀胱壁未见明显增厚, 各叶分界清晰, 肠曲分布、形态及密度未见异常, 腹膜后未见明显肿大淋巴结, 未见腹水征。增强扫描, 病灶及肾上份肾盂、肾盏呈不均匀强化, 左肾上极病灶内见不强化区。
影像结论: 左肾上极占位, 考虑肾癌可能性大。

**Ground Truth:**
影像诊断: 左肾上极可见一软组织肿物, 突出肾脏表面, 大小约43mm×38mm×36mm (左右×前后×上下), 边界清晰, CT平扫约40Hu, 增强扫描动脉期明显不均匀强化, 门脉期及延迟期强化程度减低, 右肾可见一小囊状无强化结节, 直径约10mm, 边界清晰, 双侧肾上腺形态、大小、密度未见明显异常, 双侧肾盂、肾盏未见明显扩张积水, 所见肠曲分布、形态及密度未见异常, 所见肝脏、胆囊、胰腺、脾脏形态、大小、密度未见明显异常, 腹膜后未见明显肿大淋巴结, 未见腹水征。
影像结论: 左肾上极软组织肿物, 考虑肾癌可能性大, 右肾小囊肿, 余未见其它异常。

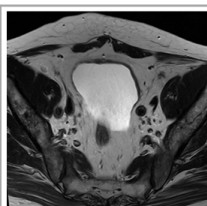

**MedRegA Prediction:**
影像诊断: 前列腺增大, 约48mm×38mm×47mm (左右径×前后径×上下径), 部分突向膀胱内, 前列腺移行带信号欠均匀, 见多发大小不等结节, 增强扫描中央腺体呈明显强化, 移行带多发结节强化, 前列腺下叶外周带见一类圆形结节, 约16mm×12mm, 呈低信号, 增强扫描呈明显强化, 动态增强扫描前列腺强化明显, 膀胱充盈, 壁未见明显增厚, 盆腔未见明显积液, 扫及骨盆骨质未见明确异常。
影像结论: 前列腺增生, 前列腺移行带多发结节, 考虑前列腺癌, 前列腺下叶外周带小血管瘤, 建议必要时进一步检查。

**Ground Truth:**
影像诊断: 前列腺体积增大, 部分突入膀胱底壁, 大小约60mm×59mm×45mm, 中央带信号不均匀, 其内可见斑片状长t1长t2信号影, 增强后前列腺强化较均匀, 前列腺外周带及移行带左侧各见一结节, 较大者大小约22mm×13mm, t2wi呈低信号, , 增强扫描未见异常强化信号, 膀胱充盈良好, 壁光整, 未见结节或肿块影, 盆腔周围肌间隙清晰盆腔未见肿大淋巴结, 扫描骨质未见明确异常信号灶。
影像结论: 前列腺增生, 前列腺外周带及移行带左侧结节, 考虑前列腺癌。

Figure 14: **Generated report examples on our In-house Chinese report dataset.**

Table 8: **Performance comparison on Chinese Report Generation task.** '-' indicates the model cannot generate valid outputs.

| Dataset | Modality | | Metrics | Med-Flamingo | LLaVA-Med | RadFM | MedDr | Ours |
|---|---|---|---|---|---|---|---|---|
| | | | *Report Generation in Chinese* | | | | | |
| Brain | MRI, CT | NLG | BLEU-1 | 5.17 | 4.89 | 5.00 | 12.31 | **40.89** |
| | | | BLEU-4 | - | - | 0.03 | 0.72 | **20.47** |
| | | | METEOR | 3.46 | 3.19 | 5.08 | 12.30 | **39.95** |
| | | | ROUGE-L | 3.54 | 3.03 | 5.04 | 8.37 | **22.78** |
| | | | BertScore | 59.12 | 58.19 | 63.79 | 63.05 | **72.71** |
| Chest | CT | NLG | BLEU-1 | 3.40 | 3.79 | 2.35 | 10.40 | **38.47** |
| | | | BLEU-4 | - | - | 0.01 | 0.20 | **18.11** |
| | | | METEOR | 2.45 | 3.25 | 3.50 | 9.38 | **35.06** |
| | | | ROUGE-L | 2.69 | 3.35 | 3.69 | 7.29 | **19.41** |
| | | | BertScore | 59.19 | 56.72 | 62.02 | 60.71 | **69.18** |
| Spine | X-ray, MRI | NLG | BLEU-1 | 3.49 | 8.21 | 6.25 | 10.7 | **44.09** |
| | | | BLEU-4 | - | - | 0.03 | 0.16 | **22.57** |
| | | | METEOR | 2.75 | 5.35 | 5.64 | 11.23 | **29.54** |
| | | | ROUGE-L | 3.38 | 4.80 | 6.81 | 8.85 | **37.70** |
| | | | BertScore | 56.61 | 58.46 | 64.17 | 61.07 | **74.41** |
| Abdomen | CT | NLG | BLEU-1 | 2.87 | 2.85 | 1.79 | 13.51 | **45.22** |
| | | | BLEU-4 | - | - | - | 0.30 | **20.67** |
| | | | METEOR | 1.99 | 2.56 | 2.40 | 9.56 | **36.07** |
| | | | ROUGE-L | 2.37 | 2.79 | 2.66 | 8.84 | **18.91** |
| | | | BertScore | 58.27 | 55.67 | 60.60 | 61.07 | **69.46** |
| Pelvis | MRI | NLG | BLEU-1 | 3.02 | 4.82 | 2.91 | 13.05 | **35.12** |
| | | | BLEU-4 | - | - | 0.01 | 0.23 | **11.86** |
| | | | METEOR | 2.31 | 3.66 | 3.64 | 11.09 | **29.54** |
| | | | ROUGE-L | 2.92 | 3.93 | 4.12 | 8.84 | **16.10** |
| | | | BertScore | 57.62 | 57.08 | 62.11 | 61.04 | **69.78** |

Table 9: **Evaluation on ReXrank Benchmark.** Bold values indicate the best results, Underlined values indicate the second-best results, and '*' indicates the best results among generalist models.

| Model | Model Type | RadCliQ-v1↓ | RadCliQ-v0↓ | BLEU↑ | BertScore↑ | SembScore↑ | RadGraph↑ |
|---|---|---|---|---|---|---|---|
| | | *MIMIC-CXR Dataset* | | | | | |
| MedVersa | Generalist | **0.692*** | **2.581*** | 0.195 | **0.518*** | 0.601 | 0.244* |
| RGRG | Specialist | 0.803 | 2.818 | **0.24** | 0.447 | 0.603 | **0.248** |
| RadFM | Generalist | 0.815 | 2.8 | 0.196 | 0.479 | 0.556 | 0.234 |
| RaDialog | Specialist | 0.97 | 3.044 | 0.175 | 0.419 | 0.545 | 0.234 |
| BioMedGPT | Generalist | 1.044 | 3.175 | 0.123 | 0.361 | 0.512 | 0.242 |
| CheXagent | Specialist | 1.137 | 3.272 | 0.102 | 0.38 | 0.494 | 0.157 |
| MedRegA (Ours) | Generalist | 0.718 | 2.634 | 0.205* | 0.496 | **0.613*** | 0.244* |
| | | *IU Xray Dataset* | | | | | |
| MedVersa | Generalist | **1.088*** | **3.337*** | **0.193*** | **0.43*** | 0.315 | **0.273*** |
| RGRG | Specialist | 1.363 | 3.723 | 0.125 | 0.323 | 0.337 | 0.176 |
| RadFM | Generalist | 1.604 | 4.093 | 0.081 | 0.281 | 0.245 | 0.111 |
| RaDialog | Specialist | 1.33 | 3.647 | 0.112 | 0.322 | **0.381** | 0.168 |
| BioMedGPT | Generalist | 1.9 | 4.554 | 0.015 | 0.163 | 0.205 | 0.062 |
| CheXagent | Specialist | 1.437 | 3.816 | 0.094 | 0.304 | 0.331 | 0.146 |
| MedRegA (Ours) | Generalist | 1.303 | 3.644 | 0.157 | 0.358 | 0.326* | 0.185 |

samples. Notably, the gap in BertScore is much smaller than BLEU and ROUGE scores, indicating that changing the target language affects styles more significantly than semantics. Figure 15 shows several case studies on the paired bilingual dataset, reflecting the impact of language transfer. The examples show that although the style may deviate, our model can still capture the significant impressions in the medical image.

### C.1.3 MEDICAL IMAGE CLASSIFICATION

We conduct experiments on both single-label classification and multi-label classification with a wide range of datasets across radiology, ultrasound, ophthalmology, and dermatology. For all the datasets, we adopt the official test set and prompt the model with the predefined label set. Since MLLMs are inclined to overfit to frequent labels in the training set, we mitigate the class-imbalanced issue by oversampling minority classes and downsampling majority classes, which critically increases the classification score. F1 scores for each dataset are reported in Table 11. For single-label classifica-

Table 10: **Evaluation on paired bilingual data for report generation.** "Translated" indicates generating and evaluating with the translated language, and "Original" means using the original language of the report.

| Method | Language | BLEU-1 | ROUGE-L | BertScore |
|---|---|---|---|---|
| *Results on English-to-Chinese dataset* | | | | |
| Translated | CN | 10.04 | 7.06 | 66.75 |
| Original | EN | 48.99 | 34.89 | 73.98 |
| *Results on Chinese-to-English dataset* | | | | |
| Translated | EN | 19.95 | 17.39 | 62.20 |
| Original | CN | 41.66 | 32.59 | 73.39 |

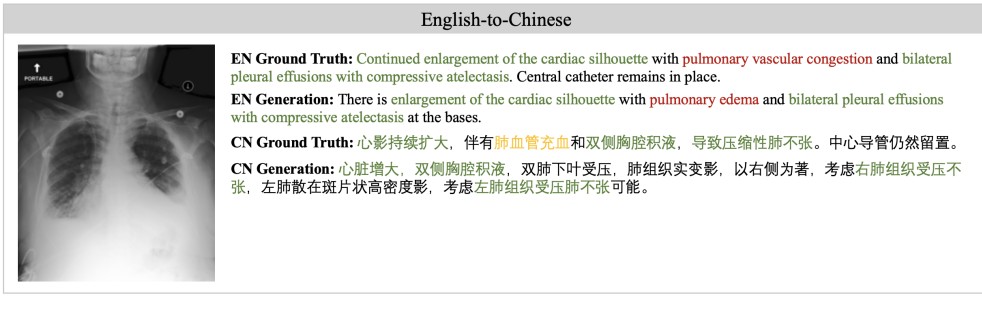

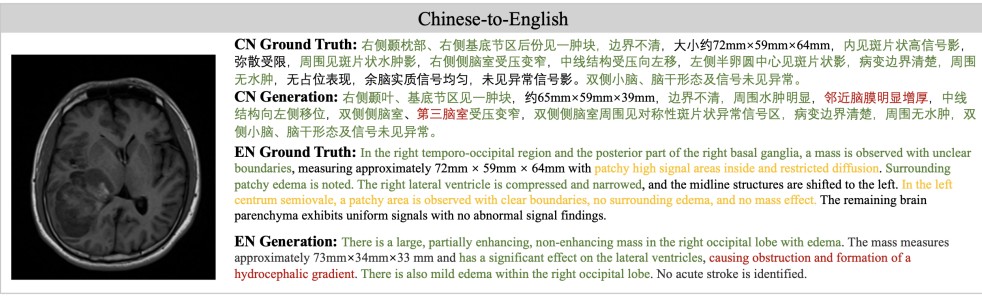

Figure 15: **Examples on paired bilingual dataset.**

tion, MedRegA outperforms existing models on most benchmarks by a large margin. In contrast, ulti-label classification appears more challenging for MLLMs due to the difficulties in decoupling subtle symptoms and relating them to corresponding diagnoses. Since Med-Flamingo, LLaVA-Med and RadFM are not exposed to diverse modalities in training, they cannot manage to generate satisfactory diagnostic results for unseen modalities, and struggle to extract valid labels from the prompts. It is worth noting that LLaVA-Med is able to acknowledge its limitation and refuse to provide uncertain answers, whereas the other models may give random responses when confronted with queries beyond their knowledge scope. Figure 16 provides examples of the generation of disease diagnosis task, demonstrating the diversity of input modalities and the diagnostic categories.

## C.2 GROUNDED REPORT GENERATION

As a supplement to the main text, we present the comparison in the quality of reports for organ grounded report generation in Table 12. NLG metrics are adopted to evaluate the report outputs. By integrating the region-centric task into prediction, our model shows further improvements in the result of report generation, highlighting the value of regional information in enhancing the performance of textual generation. Additionally, we exemplify the output of MedRegA for the grounded report generation task in Figure 17. The examples reveal that even when the previous SOTA method MedDr successfully detect the abnormality in the given medical scan, it still struggles to illustrate the specific region of the diagnosis, hindering further performance improvement and interoperability. In comparison, our proposed MedRegA is capable of locating the specific area related to each descriptive sentence, accelerating its feasibility for clinicians.

Table 11: **Performance comparison on Medical Image Classification task.** '-' indicates the model cannot generate valid outputs.

| Dataset | Modality | Med-Flamingo | LLaVA-Med | RadFM | MedDr | Ours |
|---|---|---|---|---|---|---|
| *Single-Label Classification* | | | | | | |
| MURA | X-ray | 34.30 | 36.31 | 49.50 | 37.81 | **69.99** |
| PneumoniaMNIST | X-ray | 69.72 | 57.43 | 38.02 | **87.30** | 85.23 |
| OrganAMNST | CT | 4.78 | 14.90 | 1.23 | 20.70 | **29.37** |
| OrganCMNIST | CT | 1.49 | 7.50 | 6.62 | 6.99 | **24.22** |
| OrganSMNIST | CT | 1.32 | 7.27 | 6.49 | 8.68 | **18.98** |
| ISIC2016 | Dermatology | 44.51 | 43.02 | 37.60 | 50.73 | **65.16** |
| ISIC2018 | Dermatology | 10.73 | 4.57 | 7.46 | 11.84 | **16.02** |
| PAD-UFES-20 | Dermatology | 2.59 | 12.90 | 13.81 | 14.09 | **20.24** |
| KatherColon | Pathology | 10.37 | 12.40 | 6.54 | 16.65 | **43.23** |
| Messidor-2 | Fundus | 17.79 | 29.84 | 18.31 | 39.72 | **54.74** |
| OCT2017 | OCT | 13.33 | 20.97 | 23.24 | 41.13 | **88.45** |
| OCTMNIST | OCT | 10.00 | 31.50 | 19.97 | 58.30 | **64.46** |
| BUSI | Ultrasound | 22.60 | 22.94 | 29.8 | 30.90 | **45.20** |
| BreastMNIST | Ultrasound | 26.79 | 39.72 | 27.11 | 66.10 | **70.35** |
| *Multi-Label Classification* | | | | | | |
| CheXpert | X-ray | 4.66 | 18.99 | 15.60 | 6.95 | **21.43** |
| ChestMNIST | X-ray | - | - | 4.90 | **13.40** | 11.96 |
| CXR14 | X-ray | 5.65 | 7.80 | 9.35 | 8.74 | **10.28** |
| VinDr-CXR | X-ray | 1.59 | 2.36 | 6.57 | 7.11 | **11.89** |
| VinDr-PCXR | X-ray | 2.30 | 1.92 | 6.29 | **8.20** | 6.33 |
| VinDr-SpineXR | X-ray | 3.72 | 2.49 | 14.20 | 26.80 | **30.16** |
| VinDr-Mammo | Mammography | 0.85 | 3.25 | 5.7 | 8.87 | **9.99** |
| BRSET | Fundus | 2.48 | 7.93 | 6.76 | 2.49 | **11.28** |
| RFMiD | Fundus | 1.53 | 2.65 | 5.54 | 1.89 | **6.55** |

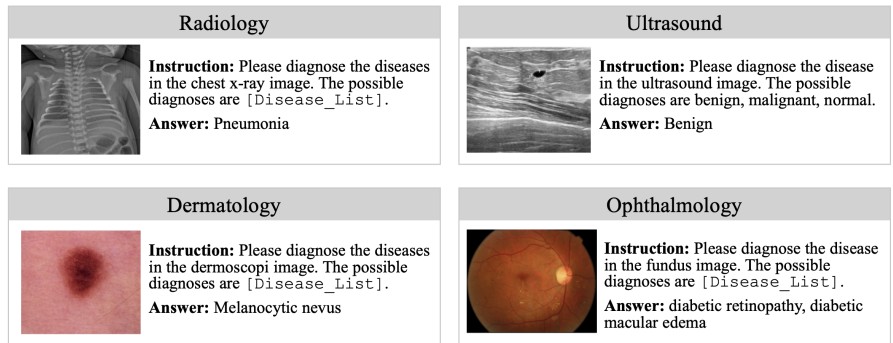

Figure 16: **Disease diagnosis examples. [Disease_List] represents all the diseases included in the corresponding dataset.**

Table 12: **Performance comparison on the quality of reports for organ grounded report generation.**

| Model | BLEU-1 | BLEU-4 | METEOR | ROUGE-1 | ROUGE-L | BertScore |
|---|---|---|---|---|---|---|
| InternVL | 10.00 | 2.54 | 7.89 | 8.90 | 7.32 | 59.20 |
| MedDr | 13.42 | 4.15 | 8.10 | 11.45 | 10.32 | 60.50 |
| MedRegA | **28.20** | **9.55** | **27.01** | **22.86** | **16.72** | **66.46** |

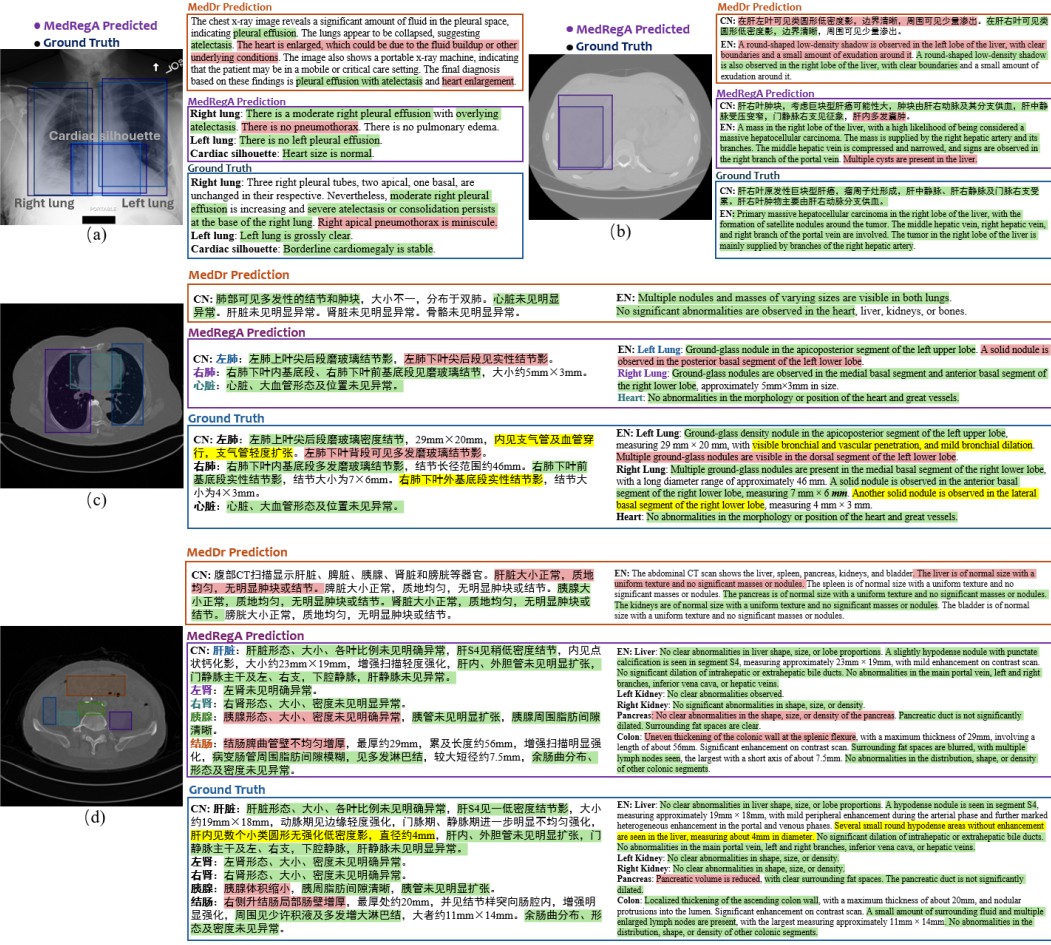

Figure 17: **Generated Grounded Report Examples.** (a) shows the example on the MIMIC-CXR-sourced grounded report dataset. (b) demonstrates the example for the grounded reports focusing on the lesion part. (c) and (d) present the examples of organ grounded reports.

## C.3 COMPARISON WITH FINE-TUNED BASELINES

We have evaluated the fine-tuning results with two baselines, Med-Flamingo (Moor et al., 2023) and LLaVA-Med (Li et al., 2024) on the region-centric tasks to prove the effectiveness and potential of our proposed dataset. We apply a subset of 5K samples from the proposed MedRegInstruct dataset to fine-tune the baseline models. This subset covers Region-to-Text Identification and Text-to-Region Detection tasks. We evaluated the models using the test split of the same subset. For Med-Flamingo (Moor et al., 2023), a few-shot learner adapted to the medical domain, we perform 5-shot learning with our dataset. For LLaVA-Med (Li et al., 2024), we finetuned the model on the sub-dataset for 10 epochs. Table 13 shows the results. It can be observed that both Med-Flamingo (Moor et al., 2023) and LLaVA-Med Li et al. (2024) can improve on the regional tasks after finetuning with our proposed dataset, demonstrating the effectiveness of our dataset in extending regional knowledge.

Table 13: **Comparision with fine-tuned baselines on Region-Centric tasks.** 'N/A' indicates the model cannot generate valid outputs

| Metric | Med-Flamingo | | LLaVA-Med | | MedRegA |
|---|---|---|---|---|---|
| | Zero Shot | Few Shot | Base | Fine-tuned | |
| *Region-to-Text Identification Task* | | | | | |
| BLEU | 2.19 | 32.02 | 0.05 | 21.49 | **59.06** |
| F1 | 3.79 | 33.19 | 0.14 | 25.08 | **59.21** |
| Recall | 11.70 | 33.40 | 1.99 | 32.23 | **59.19** |
| Accuracy | 6.62 | 16.87 | 0.91 | 31.93 | **51.72** |
| BertScore | 50.36 | 71.53 | 32.18 | 58.21 | **82.49** |
| *Text-to-Region Detection Task* | | | | | |
| Region-Level F1 | | 18.36 | | 19.68 | **56.08** |
| Alignment F1 | N/A | 20.65 | N/A | 19.53 | **52.52** |
| IoU | | 19.52 | | 25.27 | **47.43** |

# D REGION-ALIGNED EVALUATION

To quantitatively evaluate the medical MLLMs' regional perception and comprehension capabilities on Region-Centric tasks, we introduce a Region-Aligned evaluation framework to measure the model performance on these tasks.

**Region-to-Text Identification task**. Since the output for region-to-text identification task is in the form of pure texts, we adopt Natural Language Generation (NLG) metrics for performance measurement, including BLEU-1, F1 score, Recall, Accuracy and BertScore.

**Text-to-Region Detection task**. We classify thetext-to-region detection task into four categories according to the number of detected objects and the number of regions per object: single-object single-region, single-object multi-region, multi-object single-region, and multi-object multi-region, as illustrated in §A.1.1. We evaluate the performance across three dimensions: (1) *Object-Level*: assessing the correctly identified objects; (2) *Region-Level*: assessing accurately detected regions; (3) *Object-Region Alignment*: assessing whether the detected boxes are correctly aligned with the corresponding object.

For single-object detection, all the output boxes are aligned with the given object, where evaluating the detection performance from the region-level is sufficient. The region-level evaluation metrics are represented in Figure 18. If the object is related to only one region, we report the accuracy of the detected boxes. However, if the object corresponds to multiple regions, we calculate the precision, recall and F1 score.

For multi-object detection, the alignment of text and box must also be considered. As illustrated in Algorithm 1, we first apply Hungarian Matching to get an optimal IoU-based assignment for object-region pairs set of each input, where each predicted region is matched with the ground truth with the maximum IoU score. Subsequently, for each matched predicted and ground truth object-region pair, we identify whether the object and region are correctly detected andproperly ligned. We calculate the three-dimensional metrics for each sample and utilize the averaged score for overall evaluation.

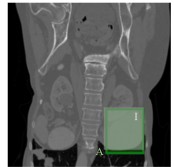 Singe-object Singe-region

Region-Level Accuracy=$\frac{\text{\# Detected Regions}}{\text{\# All the Regions}}$

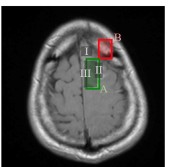 Singe-object Multi-region

Region-Level Prediction=$\frac{\text{\# Detected Regions}}{\text{\# All the Regions}}$=1/3

Region-Level Recall=$\frac{\text{\# Detected Regions}}{\text{\# Predicted Regions}}$=1/2

Figure 18: **Region-level Evaluation for Single-Object Detection.**

---

**Algorithm 1** Region-Aligned Evaluation

---

**Input:**
    Predicted Object-Region Pairs Sets $\mathcal{P}$ for all the inputs
    Ground Truth Object-Region Pairs Sets $\mathcal{G}$ for all the inputs
**Output:**
    Region-Aligned Evaluation Metrics
**for** each $P_i = \{(p_n^{text}, p_n^{region})\}^N \in \mathcal{P}, G_i = \{(g_m^{text}, g_m^{region})\}^M \in \mathcal{G}$ **do**
    Optimal IoU Assignment $\{(p_n^{region}, g_m^{region})\}^{\min(N,M)} \leftarrow Hungarian\_Matching(P_i, G_i)$
    **for** each $(p_n^{region}, g_m^{region})$ **do**     ▷ iterate over each matched object-region pair for an input
        Detected Objects $\leftarrow 0$
        Detected Regions $\leftarrow 0$
        Aligned Pairs $\leftarrow 0$
        Calculate IoU between $(p_n^{region}$ and $g_m^{region})$
        **if** $p_n^{text} = g_m^{text}$ **then**     ▷ the object is correctly detected
            Detected Objects $\leftarrow$ Detected Objects + 1
        **end if**
        **if** IoU $\geq 0.5$ **then**     ▷ the region is correctly detected
            Detected Regions $\leftarrow$ Detected Regions + 1
        **end if**
        **if** $p_n^{text} = g_m^{text}$ & IoU $\geq 0.5$ **then**     ▷ current object-region pair is correctly aligned
            Aligned Pairs $\leftarrow$ Aligned Pairs + 1
        **end if**
    **end for**
    Object-level Precision $\leftarrow \frac{\text{Detected Objects}}{\text{All the Objects}}$, Object-level Recall $\leftarrow \frac{\text{Detected Objects}}{\text{Predicted Objects}}$
    Region-level Precision $\leftarrow \frac{\text{Detected Regions}}{M}$, Region-level Recall $\leftarrow \frac{\text{Detected Regions}}{N}$
    Object Region-level Alignment Precision $\leftarrow \frac{\text{Aligned Pairs}}{M}$
    Object Region-level Alignment Recall $\leftarrow \frac{\text{Aligned Pairs}}{N}$     ▷ calculate the metrics for each sample
**end for**
Calculate the averaged metrics over all the inputs

---

**Grounded Report Generation task**. The evaluation of Grounded Report Generation performance is composed of textual evaluation and regional evaluation. For the textual evaluation, we apply NLG metrics to assess the quality of the report part, while regional evaluation follows the same discipline as text-to-region detection task.

## E   EFFECTIVENESS OF REGIONAL CoT

Since our model has the ability to perceive regions, we also explore the impact of guiding the model with abnormal regions to enhance generation. For instance, we test the model on the visual question answering dataset SLAKE and multi-label classification VinDr Series datasets, by enforcing the model to first identify abnormal areas and then prompt the model with the bounding box. Table 14 indicates that providing guidance with abnormal region greatly improves the performance, essentially on the challenging multi-label classification task (with 61.75% and 19.74% F1 score on VinDr-SpineXR and VinCXR dataset, respectively), proving the immense value of understanding regional knowledge.

Table 14: **Effectiveness of our proposed Regional CoT.**

| Dataset | Metrics | MedDr | w/o Regional CoT | w/ Regional CoT |
|---|---|---|---|---|
| VinDr-CXR | F1 | 7.11 | 11.89 | **19.74** |
| VinDr-PCXR | F1 | 8.20 | 6.33 | **16.49** |
| VinDr-SpineXR | F1 | 26.80 | 30.16 | **61.75** |
| VinDr-Mammo | F1 | 8.87 | 9.99 | **12.91** |
| SLAKE | BLEU-1 | 76.40 | 81.55 | **84.61** |
| | F1 | 77.50 | 82.06 | **85.32** |
| | CloseAccuracy | 83.40 | 85.35 | **87.89** |
| | OpenRecall | 74.20 | 80.45 | **83.61** |

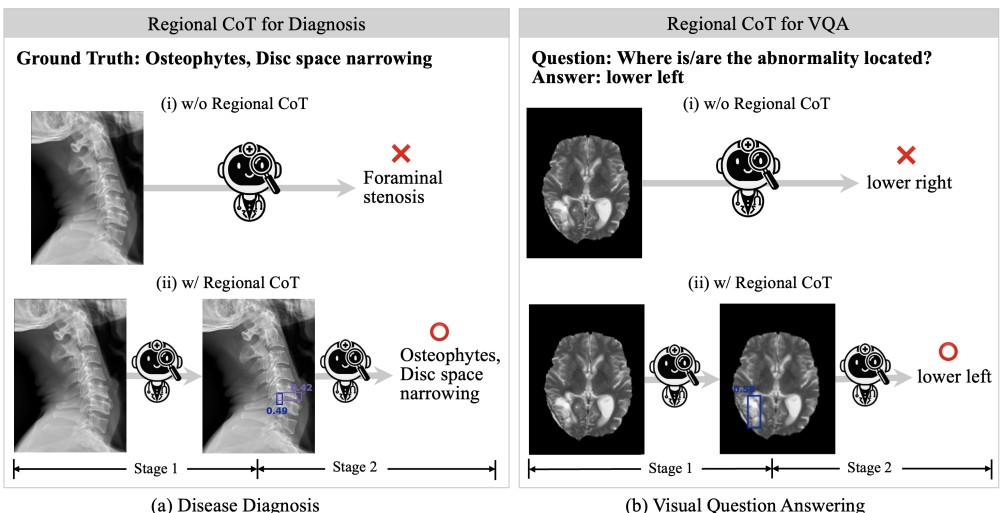

Figure 19: **Ablation Study of Regional CoT.**

Additionally, we present Regional CoT examples as an ablation study to further prove its effectiveness in Figure 19. As shown in Figure 19 (a) for disease diagnosis task, the model struggles to directly predict the correct disease but successfully identifies all relevant diseases with Regional CoT. This can be attributed to the detection stage in Regional CoT, which encourages the model to first detect the abnormal regions, thereby mimicking the diagnostic process of clinicians. Figure 19 (b) also highlights the impact of Regional CoT on VQA tasks, especially in enhancing the model's awareness of specific locations.

