# OpenReview forum: "Interpretable Bilingual Multimodal Large Language Model for Diverse Biomedical Tasks"
_ICLR.cc/2025/Conference — ICLR 2025 Poster_

### Official Review · Reviewer_WceT · 2024-10-31

**Soundness:** 4
**Presentation:** 4
**Contribution:** 4
**Rating:** 6
**Confidence:** 4

**Summary:**

This paper presents MedRegA, a region-aware medical multimodal language model capable of handling both image-level and region-level vision-language tasks across diverse medical modalities. The authors propose three Region-Centric tasks and create a large-scale dataset called MedRegInstruct to enhance the model's focus on specific anatomical regions. They introduce a Regional Chain-of-Thought approach to further improve performance. Experiments show that MedRegA outperforms existing models on tasks like visual question answering, report generation, and medical image classification, demonstrating significant versatility and interpretability.

**Strengths:**

- The comprehensive MedRegInstruct dataset is valuable and likely to have a significant impact on the community.
- Introducing Regional Chain-of-Thought is insightful and significantly improves the model's performance.
- The model achieves state-of-the-art performance without fine-tuning on specific benchmark datasets, indicating strong generalization across different domains.

**Weaknesses:**

- Human validation of the automatically generated annotations is needed to prove the quality of the dataset.
- The paper lacks comprehensive comparisons in report generation with state-of-the-art methods, such as those on the ReXrank[1] leaderboard.
- Detection task of multiple regions still remain challenging for purposed method.

[1] Rajpurkar Lab. ReXrank Leaderboard. 22 July 2024, https://rajpurkarlab.github.io/rexrank/.

**Questions:**

Please address the weaknesses mentioned above.

---

> ### Author Response · Authors · 2024-11-21
> **Thanks for your precious comments. [Part I]**
>
> We are immensely grateful to you for recognizing the importance and novelty of our work. Thanks for your precious comments.
>
> **W1: "Human validation of the automatically generated annotations is needed to prove the quality of the dataset."**
>
> Thanks for your suggestion. We have conducted human validation for the automatically annotated data, as detailed in the Common Questions Q2.
>
> **W2: "The paper lacks comprehensive comparisons in report generation with state-of-the-art methods, ..."**
>
> Thanks for your advice. We have performed additional comparisons in report generation on ReXrank with official splits and metrics. The tables below demonstrate the results.
>
> **Results on IU-Xray benchmark.**
>
> | **Model**          | **Model Type** | **RadCliQ-v1↓** | **RadCliQ-v0↓** | **BLEU↑** | **BertScore↑** | **SembScore↑** | **RadGraph↑** |
> | ------------------ | :------------: | :-------------: | :-------------: | :-------: | :------------: | :------------: | :-----------: |
> | MedVersa           |   Generalist   |   **0.692**    |   **2.581**    |   0.195   |   **0.518**   |     0.601      |    0.244*     |
> | RGRG               |   Specialist   |      0.803      |      2.818      | **0.24**  |     0.447      |     0.603*      |   **0.248**   |
> | RadFM              |   Generalist   |      0.815      |       2.8       |   0.196   |     0.479      |     0.556      |     0.234     |
> | RaDialog           |   Specialist   |      0.97       |      3.044      |   0.175   |     0.419      |     0.545      |     0.234     |
> | BioMedGPT          |   Generalist   |      1.044      |      3.175      |   0.123   |     0.361      |     0.512      |     0.242     |
> | CheXagent          |   Specialist   |      1.137      |      3.272      |   0.102   |      0.38      |     0.494      |     0.157     |
> | **MedRegA (Ours)** |   Generalist   |      0.718*      |      2.634*      |  0.205*   |     0.496*      |   **0.613**   |    0.244*     |
>
> **Results on MIMIC-CXR benchmark.**
>
> | **Model**          | **Model Type** | **RadCliQ-v1↓** | **RadCliQ-v0↓** | **BLEU↑**  | **BertScore↑** | **SembScore↑** | **RadGraph↑** |
> | ------------------ | :------------: | :-------------: | :-------------: | :--------: | :------------: | :------------: | :-----------: |
> | MedVersa           |   Generalist   |   **1.088**    |   **3.337**    | **0.193** |   **0.43**    |     0.315      |  **0.273**   |
> | RGRG               |   Specialist   |      1.363      |      3.723      |   0.125    |     0.323      |     0.337*      |     0.176     |
> | RadFM              |   Generalist   |      1.604      |      4.093      |   0.081    |     0.281      |     0.245      |     0.111     |
> | RaDialog           |   Specialist   |      1.33       |      3.647      |   0.112    |     0.322      |   **0.381**    |     0.168     |
> | BioMedGPT          |   Generalist   |       1.9       |      4.554      |   0.015    |     0.163      |     0.205      |     0.062     |
> | CheXagent          |   Specialist   |      1.437      |      3.816      |   0.094    |     0.304      |     0.331      |     0.146     |
> | **MedRegA (Ours)** |   Generalist   |      1.303*      |      3.644*      |   0.157*    |     0.358*      |     0.326     |     0.185*     |
>
> In the table, **Bold** values indicate the best results, and '*' indicates the second-best results.
>
> While our model's performance on certain metrics does not surpass all the models on the ReXrank benchmark, we believe the following points highlight the contribution of our approach:
>
> 1. **Comparable Results with Specialist Models:** Our model achieves results that are comparable to specialist models, such as RGRG [2] and RaDialog [3], which are specifically trained on report generation tasks for chest X-ray datasets. However, unlike these specialist models, our model is designed as a generalist framework capable of handling multiple tasks, such as VQA and diagnosis, and supporting more modalities.
> 2. **Competitive Performance Among Generalist Models:** Within the generalist model category, our model ranks second. It accomplishes superior scores in certain metrics compared to MedVersa [1], and outperformed other generalists including RadFM [4] and BioMedGPT [5]. In comparison with MedVersa [1], our model offers broader versatility in report generation for different modalities of various organs, including X-ray, CT and MRI scans of the brain, spine, abdomen, and pelvis, in addition to chest X-rays.
> 3. **Contributions Beyond Benchmark Performance**: While this benchmark focuses solely on chest X-ray report generation, our model introduces unique capabilities for region-specific tasks.
>
> We have updated the results on ReXrank benchmark in **Table 9 of Appendix C.1.2**.

---

> > ### Author Response · Authors · 2024-11-21
> > **Thanks for your precious comments. [Part II]**
> >
> > **W3: "Detection task of multiple regions still remain challenging for purposed method."**
> >
> > Thanks for your careful review. We acknowledge that detecting multiple regions simultaneously remains challenging compared with single-region detection. This is possibly due to the complex relationships between the input texts with multiple corresponding regions, which requires the language model to handle detection and alignment at the same time. Despite the challenges, we believe our work established a strong foundation for multi-region detection in medical images using MLLM with the proposed region-centric dataset and region-aware model. The performance on region detection tasks also demonstrates the feasibility of our approach. In the future, we will explore to further improve the performance of multi-region detection for medical images, for example, by incorporating specific detection or segmentation models with our model.
> >
> >
> > ### **Reference**
> > [1] Zhou, Hong-Yu, et al. "A Generalist Learner for Multifaceted Medical Image Interpretation." arXiv preprint arXiv:2405.07988 (2024).
> >
> > [2] Tanida, Tim, et al. "Interactive and explainable region-guided radiology report generation." Proceedings of the IEEE/CVF Conference on Computer Vision and Pattern Recognition. 2023.
> >
> > [3] Pellegrini, Chantal, et al. "RaDialog: A large vision-language model for radiology report generation and conversational assistance." arXiv preprint arXiv:2311.18681 (2023).
> >
> > [4] Wu, Chaoyi, et al. "Towards generalist foundation model for radiology." arXiv preprint arXiv:2308.02463 (2023).
> >
> > [5] Zhang, Kai, et al. "A generalist vision–language foundation model for diverse biomedical tasks." Nature Medicine (2024): 1-13.

---

> > > ### Comment · Reviewer_WceT · 2024-11-23
> > > **Thanks for your response.**
> > >
> > > The authors have adequately addressed all of my concerns. The results in human validation and report generation are promising, suggesting that this dataset has significant potential to benefit the broader research community.
> > >
> > > After reviewing the rebuttal and considering other reviews, I have decided to raise my rating to 7.

---

### Official Review · Reviewer_ToYk · 2024-11-01

**Soundness:** 3
**Presentation:** 3
**Contribution:** 3
**Rating:** 6
**Confidence:** 5

**Summary:**

This work introduces a novel region-centric task and presents a large-scale Chinese medical dataset named MedRegInstruct. To address this new task, the authors developed a bilingual multimodal large language model, MedRegA, which outperforms baseline methods across most evaluation metrics.

**Strengths:**

1.	The paper is well-organized.
2.	This work introduces a novel task alongside a large-scale medical dataset.
3.	The authors propose a new model that significantly outperforms other models across the majority of datasets and evaluation metrics, particularly for the newly proposed task.

**Weaknesses:**

1.	The article’s description of the model structure in the main text is incomplete, relying heavily on Figure 4 to convey the model’s architecture. For example, the structure of the encoder and tokenizer remains ambiguous. The authors should clarify which components utilize existing models, which are fine-tuned, and which are trained from scratch.
2.	The fairness of the model comparison is questionable. In Table 1, only a subset of models has been fine-tuned based on the test dataset, leaving others unmodified. Is this due to an inability to fine-tune certain models, or for another reason? The authors should provide detailed criteria for choosing to fine-tune specific models and address the fairness of this experimental design.
3.	On the newly proposed task, the authors’ model demonstrates a substantial performance advantage. They mention that existing open-source models cannot accurately capture coordinates, but it would be insightful to know if fine-tuning on the authors’ proposed dataset could improve this. If fine-tuning other models would yield better results, the authors should do so for a more persuasive comparison. Conversely, if fine-tuning has no impact, the authors should explain the model’s unique advancements to provide deeper insights.
4.	The role of the CoT component is somewhat unclear, with insufficient ablation experiments to support its effectiveness. The authors should conduct a more specific ablation study to illustrate the impact of CoT rather than merely reporting improved results with its inclusion.
5.	While the authors claim that the proposed model can perform diagnostic tasks, there is no clear evaluation or user study provided. A brief evaluation or explanation of the diagnostic results would add clarity.
6.	Several prior studies focus on medical image segmentation, which aligns well with the proposed region-centric task. Have the authors considered using such methods as feature extractors, backbones, or even as comparative models?
7.	The experimental results on report generation are insufficient, as the authors rely solely on descriptive analysis without quantitative evaluation or user studies. Similar to point 5, if the authors wish to validate effectiveness without explicit evaluation metrics, a user study would provide valuable support.

**Questions:**

1.	Please explain why certain models were fine-tuned in the comparison while others were not.
2.	The authors state that adding CoT improves model performance; however, no comprehensive ablation experiments were conducted to confirm this.
3.	Have the authors considered comparisons using other large models as the backbone?
4.	Do the authors plan to open-source this dataset?

---

> ### Author Response · Authors · 2024-11-21
> **Thanks for your time in the review process and the insightful comments.**
>
> We are grateful for the reviewer's appreciation of our model and the paper's clarity. Thanks for your time in the review process and the insightful comments.
>
> ## Weaknesses
>
> **W1: "The article’s description of the model structure in the main text is incomplete,..."**
>
> Thanks for your advice. We revised the main text in Section 4 and included additional training details.
>
> **W2: "The fairness of the model comparison is questionable..."**
>
> Thanks for the question. Certain models (LLaVA-Med [1] and BioMedGPT [2]) were marked as "fine-tuned" because the checkpoints provided have been fine-tuned on specific datasets. For LLaVA-Med [1], the fine-tuned models on VQA benchmarks, SLAKE, VQA-RAD, PathVQA, are open-sourced. For BioMedGPT [2], the fine-tuned results of these three VQA benchmarks are reported in the original paper. Although fine-tuning on targeted tasks before evaluation can often enhance performance, the fine-tuning process may reduce deployment efficiency. Our MedRegA model does not require the fine-tuning stage, and outperforms the finetuned models on overall results in a zero-shot manner.
>
> [1] Li, Chunyuan, et al. "Llava-med: Training a large language-and-vision assistant for biomedicine in one day." Advances in Neural Information Processing Systems 36 (2024).
>
> [2] Zhang, Kai, et al. "A generalist vision–language foundation model for diverse biomedical tasks." Nature Medicine (2024): 1-13.
>
> **W3: "...it would be insightful to know if fine-tuning on the authors’ proposed dataset could improve this..."**
>
> Thanks for your suggestions. We would like to clarify that our proposed MedRegInstruct dataset is a core contribution, as it incorporates region-centric ability into medical MLLMs, which has not been explored in previous research. Specifically, we applied an automatic labeling system to curate the grounded medical reports in the dataset. Our dataset covers comprehensive region-centric tasks, aimed at encouraging the model to focus on critical areas. Besides, these baselines were all trained using different data sources and different data construction methods, making direct comparisons challenging. Notably, the design of dataset construction is one of the key contributions to the development of MLLM.
>
> Moreover, we have evaluated and analyzed the fine-tuning results on our proposed dataset with two baselines, Med-Flamingo and LLaVA-Med, to demonstrate the effectiveness and novelty of our proposed dataset, as demonstrated in the Common Questions Q1.
>
> **W4: "The role of the CoT component is somewhat unclear, ..."**
>
> Thanks for your suggestion. We have presented a more specific ablation study in **Appendix E** to further demonstrate the effectiveness of regional CoT.
>
> **W5: "While the authors claim that the proposed model can perform diagnostic tasks, ..."**
>
> Thanks for the advice. We have included detailed evaluation results and user studies for diagnostic tasks in **Appendix C.1.3** to demonstrate the model performance on diagnostic tasks.
>
> **W6: "Several prior studies focus on medical image segmentation, ..."**
>
> Thanks for your careful review. We agree that some segmentation models, like MedSAM [1], possess regional knowledge that helps capture the position of structures and lesions in medical images. However, these models lack training on paired image-text data, which limits their ability to align with textual representations. As a result, they are not effective on other clinically significant tasks, such as generating medical reports. While segmentation models could be used as backbones, this would require additional alignment training with image-text pairs, which is a time-consuming process that could also deviate the original segmentation semantics. Therefore, we use a multi-modal large language model as our backbone, as it already incorporates aligned image-text knowledge.
>
> [1] Ma, Jun, et al. "Segment anything in medical images." Nature Communications 15.1 (2024): 654.
>
> **W7: "The experimental results on report generation are insufficient, ..."**
>
> Thanks for the suggestions. We further provided user studies and result analysis in **Appendix C.1.2** to validate the model on the report generation task.
>
> ## Questions
>
> **Q1: "Please explain why certain models were fine-tuned in the comparison while others were not."**
>
> Thanks for your comment. Please refer to W2 for a detailed response.
>
> **Q2: "The authors state that adding CoT improves model performance..."**
>
> Thanks for your careful review. Please refer to W4 for a detailed response.
>
> **Q3: "Have the authors considered comparisons using other large models as the backbone?"**
>
> Thanks for your question. Please refer to W3 for a detailed response.
>
> **Q4: "Do the authors plan to open-source this dataset?"**
>
> Thanks for your question. The MedRegInstruct dataset will be released to the public.

---

> > ### Comment · Reviewer_ToYk · 2024-11-25
> >
> > Thank the reviewer for the detailed classification. It addressed my concerns. I have decided to raise my rating to 6.

---

### Official Review · Reviewer_5dSL · 2024-11-01

**Soundness:** 3
**Presentation:** 3
**Contribution:** 4
**Rating:** 8
**Confidence:** 4

**Summary:**

In this work, the authors tackle the current problem in MLLMs (multimodal large language models) of how they don't necessarily "focus" on particular regions, but rather take the entire image as context and then have to solve a variety of downstream tasks. To address this problem they offer two main sets of contributions. The first, is the formulation of several region specific tasks, namely region-to-text identification, text-to-region detection, and grounded report generation, which they combine into a dataset called MedRegInstruct. The second is the training strategy and modeling paradigm for a MLLM they propose, which they refer to as MedRegA. MedRegA is a bilingual, as it was trained on both English and Chinese paired datapoints, model that they apply to this larger dataset they prepare in comparison to currently available medical MLLMs.

**Strengths:**

• The paper really introduces another method of interaction, in the bounding boxes, that can be quite useful as MedRegA is able to reason about specific regions. This is an immensely useful tool for narrowing of problems as opposed to current trained MLLMs have to do.

• The proposed training strategy for learning how to incorporate the bounding boxes is simple and scalable, seems like it could be generally applied to other strategies that want to apply bounding boxes.

• Regional CoT makes a lot of sense intuitively and provides a nice bias for solving region specific problems.

• MedRegA has very strong performance on the chosen set of benchmarks and metrics in comparison to existing state of the art methods.

**Weaknesses:**

• The section of the report grounding dataset is quite confusing as to how its constructed, and how much it relies on the models it uses for automation for providing correct labels/groundtruth. For example, is InternLM perfect at sub-select different parts of the reports?

• The paper claims to work across a broad set of modalities, but when considering the grounding dataset it seems that it only considers, as other MLLMs do, Chest Xrays?

• It's unclear if the comparison to existing baselines are exactly fair as the trained MedRegA has access to more data. In fact, compared to the baseline of InternLM, it should be strictly better because MedRegA is trained with InternLM as an initialization. While it's true that the baselines don't provide a mechanism for highlighting particular regions, they still should be adapted to the new data that the authors are evaluating on.

**Questions:**

• What was the filtering criterion for reducing SA-Med2D-20M?

• Is there a noticeable performance difference between English and Chinese tasks? Is there paired data that this can be evaluated on? • • •

• Overall, it is unclear from the paper why exactly these two particular languages are used, other than being able to source data for both, rather than just arbitrary languages with medical report data.

• Is the MedRegInstruct dataset planning to be release?
For the Region-Text dataset, why is the case that there needs to be this split into half for region-to-text and half for text-to-region? Can't every datapoint server the purpose of being either during training?

---

> ### Author Response · Authors · 2024-11-21
> **Thanks for your efforts in the review and valuable comments. [Part I]**
>
> We are grateful for your recognition of our paper's soundness and novelty. Thanks for your efforts in the review and valuable comments.
>
> ## Weaknesses
>
> **W1: "The section of the report grounding dataset is quite confusing as to how its constructed..."**
>
> Thanks for your questions. We would like to further illustrate the construction of report grounding dataset.
>
> **To construct report grounding dataset for Chest Xrays in MIMIC-CXR dataset**, no automated process is required since all the labels can be retrieved in the MIMIC-CXR and Chest ImaGenome datasets.
>
> **To construct report grounding dataset for our in-house data**, we implemented two automated processes: (1) we employed InternLM to segment each report into detailed descriptions for each organ; (2) We first fine-tuned an MLLM based on the Region-Text dataset to enable it to generate regions based on organ names. Then, we input the collected images into the finetuned MLLM, and prompted it with the organs mentioned in the reports to obtain the corresponding bounding boxes. The automatically constructed part makes up approximately 10% of all the report grounding data. We also present the result of the human validation for generated data in the Common Questions Q2.
>
> **W2: "The paper claims to work across a broad set of modalities, ..."**
>
> Thanks for your question. In addition to Chest X-rays from the MIMIC-CXR public dataset, we include report grounding data constructed from our in-house data, covering more body structures such as lung, liver, and pancreas, across MRI and CT modality.
>
> **W3: "It's unclear if the comparison to existing baselines are exactly fair as the trained MedRegA has access to more data..."**
>
> Thanks for your advice. We would like to clarify that our proposed MedRegInstruct dataset is a core contribution, as it incorporates region-centric ability into medical MLLMs, which has not been explored in previous research. Specifically, we applied an automatic labeling system to curate the grounded medical reports in the dataset. Our dataset covers comprehensive region-centric tasks, aimed at encouraging the model to focus on critical areas. Besides, these baselines were all trained using different data sources and different data construction methods, making direct comparisons challenging. Notably, the design of dataset construction is one of the key contributions to the development of MLLM.
>
> Moreover, we have evaluated the fine-tuning results on our proposed dataset with two baselines, Med-Flamingo and LLaVA-Med, to demonstrate the effectiveness and novelty of our proposed dataset, as demonstrated in the Common Questions Q1.

---

> > ### Author Response · Authors · 2024-11-21
> > **Thanks for your efforts in the review and valuable comments. [Part II]**
> >
> > ## Questions
> >
> > **Q1: "What was the filtering criterion for reducing SA-Med2D-20M?"**
> >
> > Thanks for your question. We filter SA-Med2D-20M to reduce the redundancy in scans and keep scans with larger areas of interest. The original SA-Med2D-20M dataset contains multiple similar 2D slices extracted from the same 3D volume. To avoid frequently being exposed to similar samples, we filtered similar scans from the original dataset by selecting images at 50 intervals within the same scan. Besides, we removed images with a mask area smaller than 1% of the total size.
> >
> > **Q2: "Is there a noticeable performance difference between English and Chinese tasks?..."**
> >
> > Thanks for your careful review. We further apply GPT-4o to create an English-to-Chinese dataset with IU-Xray and a Chinese-to-English dataset with our in-house data. Both datasets contain 100 samples respectively. Our evaluation results on the paired bilingual data is presented below.
> >
> > **Results on English-to-Chinese dataset**
> >
> > |                | **Language** | **BLEU_1** | **ROUGE_L** | **BertScore** |
> > | -------------- | :----------: | :--------: | :---------: | :-----------: |
> > | **Translated** |      CN      |   10.04    |    7.06     |     66.75     |
> > | **Original**   |      EN      |   48.99    |    34.89    |     73.98     |
> >
> > **Results on Chinese-to-English dataset**
> >
> > |                | **Language** | **BLEU_1** | **ROUGE_L** | **BertScore** |
> > | -------------- | :----------: | :--------: | :---------: | :-----------: |
> > | **Translated** |      EN      |   19.95    |    17.39    |     62.20     |
> > | **Original**   |      CN      |   41.66    |    32.59    |     73.39     |
> >
> > In the table, ”Translated” indicates generating and evaluating with the translated language, and “Original” means using the original language of the report.
> >
> > The results demonstrate that directly prompting MedRegA to generate report in a different language results in lower performance metrics. This can be attributed to the differences in writing style between the English and Chinese report samples. Notably, the gap in BertScore is much smaller than BLEU and ROUGE scores, indicating that **changing the target language affects styles more significantly than semantics**. We have incorporated your valuable insights and further added some case studies in **Figure 15 of Appendix C.1.2** to analyze the impact of language transfer. The examples prove that although the style might deviate, our model can still capture significant impressions in the medical image.
> >
> > **Q3: "Overall, it is unclear from the paper why exactly these two particular languages are used..."**
> >
> > Thanks for your question. We will further clarify our motivation for using English and Chinese data.
> >
> > **Large amount of available data.** We use data in English and Chinese languages due to the large amount of available data, which is more easily accessible and scalable for training large models in the current stage.
> >
> > **Potential for extension to multilingual setting.** Our work provides a feasible example to achieve bilingual generation with the medical multimodal large language model, which demonstrates the potential to extend to more languages. Our experiments and case studies show that the model can learn medical knowledge effectively from both English and Chinese. Built on this success, we plan to collect data in more languages and further expand the model’s applicability in the future.
> >
> > **Q4: "Is the MedRegInstruct dataset planning to be release?..."**
> >
> > Thanks for your questions. (1) The dataset will be released. (2) We would like to further clarify the construction process of the Region-Text dataset. Instead of splitting the image into half for region-to-text and half for text-to-region, each data point is utilized for both tasks with different instructions. We ensure a balanced total data quantity for these two tasks.

---

> ### Comment · Reviewer_5dSL · 2024-11-25
> **Response to authors.**
>
> Thank you to the authors for the in-depth response my concerns and questions, as well as clearing up a few points I was confused with. I agree that the MedRegInstruct dataset will be a valuable contribution to the research community, and that with the additional results on attempting to tune the baselines for fair comparison that I am raising my score to an 8.

---

### Official Review · Reviewer_QLEK · 2024-11-02

**Soundness:** 3
**Presentation:** 3
**Contribution:** 4
**Rating:** 8
**Confidence:** 4

**Summary:**

This paper proposes a new large-scale, multi-site, medical visual-question answering (VQA) dataset, called MedRegInstruct, with fine-grained region-specific annotation and multiple corresponding region-centric tasks. The proposed dataset contains multiple region-centric evaluation tasks including, region-to-text generation, text-to-region detection, and grounded medical report generation. The paper further proposed a new region-aware multi-modal large language model (MLLM), called, MedRegA, pre-trained with the proposed dataset and regional chain of thought (CoT). The proposed method outperforms multiple existing SOTA medical VQA models on the regional-centric evaluation.

**Strengths:**

1. The proposed new region-centric medical dataset MedRegInstruct addressed the need for fine-grained, region-level, training and evaluation datasets in the medical domain. It is novel to the field and will help the development of the field as provides a more fine-grained training and evaluation annotation. It is proven that the method pre-trained with the new dataset can demonstrate more robust performance in the targeted region-specific tasks and also other related medical tasks.
2. The paper has provided enough details about the data curation and evaluation, including the process of question-answering pair generation from medical data with region annotation, a prompt for each downstream evaluation, and detailed settings for each task. The author also seems to claim they will release the code and pre-trained model later.
3. The paper is overall well written with nice figures and a clean presentation. It is easy to follow even if there are many details in it.

**Weaknesses:**

1. While the proposed dataset is novel and important to the field, the corresponding MedRegA model is not that novel, though I hate to say that, in comparison. The overall model design is basically the same as the base InternVL model and the major change is the training scheme and prompt formulation with the new dataset, which can be extended to other medical VQL models as well. While the author further proposed single-step regional CoT in the paper, it is still relatively straightforward. **Yet**, this is not a major drawback as the performance of this model demonstrates the superiority of training with the proposed dataset.
2. Similarly, it seems that the baselines used in the evaluation are not all fine-tuned with the proposed dataset, which enables the model to understand regional information. Comparing the model trained on the proposed dataset with baselines that are not trained on this dataset is not very fair, to some extent. It would be interesting to see how well each baseline would improve if they were fine-tuned on the proposed dataset, even if just in a few-shot manner.
3. Another concern about this paper is the quality of the data. Most of the regional bounding box-text pairs in the dataset were taken from the existing dataset and generated from an automated rule-based system and an LLM. The evaluation of the data quality is missing in this paper, which is very critical as the main purpose is **fine-grained regional evaluation,** where the quality of the bounding box and the correctness of the corresponding text description are very important. It would be better if some sort of data quality check could be done before releasing it to the public.

**Questions:**

Overall, this is a good paper with solid contributions and it has a significant meaning to the field. Still, the reviewer has a few questions here.

1. How is the English version of the text report/caption generated for the in-house Chinese data? What kind of translation was used here?
2. According to the “Structure Detection” section on page 5, the structure bounding box in the dataset was generated with a fine-tuned MLLM on the existing Region-Text dataset given the corresponding organ prompt. The reviewer wonders if any kind of evaluation was done on this fine-tuned MLLM, how accurate is the extract bounding box? And why not train a more straightforward detection model for this purpose? Considering that the specific detection model usually performs better than a general MLLM.
3. Also, as mentioned above, the reviewer wonders how much will the other baseline medical MLLMs improve if they were trained with the proposed dataset. The comparison between InternVL and MedRegA in the paper is not very fair as the InternVL was only trained with natural image-text pairs. The reviewer understands it may take tons of extra time/money to conduct such an evaluation, but it would still be an interesting question to explore.
4. According to the abstract, the code and model will be released to the public, but the reviewer wonders if the data will be released as well. This is pretty critical as the dataset is the major contribution of the paper. The answer to this question will influence the reviewer's final score for the paper.

**Details Of Ethics Concerns:**

The paper proposed a new medical visual question-answering dataset with in-house medical imaging data from a Chinese hospital. Though the author has an ethics statement section in the paper, further ethics evaluation may be needed to ensure there is no critical ethical issue.

---

> ### Author Response · Authors · 2024-11-21
> **Thanks for your constructive comments.**
>
> We are deeply grateful for your recognition of our paper's performance and potential academic impact. Thanks for your constructive comments.
>
> ## Weaknesses
>
> **W1: "While the proposed dataset is novel and important to the field, ..."**
>
> Thanks for your careful review. We appreciate your recognition of the significance of our proposed dataset. We would like to highlight and clarify the contributions of our research.
>
> **Our key contribution lies in enhancing the region-centric ability into the medical generalist models, which remains unexplored in previous works.** Considering the limitation of existing works in detecting or reasoning about specific regions within a medical scan, we contributed to enhancing spatial awareness of medical generalist models and performing specific adaptation on region-centric tasks. This significantly improves the model's ability to generate high-quality and interpretable outputs in complex medical scenarios. Moreover, the proposed regional CoT strategy enables interpretable and region-specific reasoning, inspired by the diagnostic process of clinicians who rely on precise and localized information for diagnosis. By incorporating this strategy, the regional knowledge within our model can be leveraged to further enhance downstream tasks like VQA and diagnosis, demonstrating stronger potential in clinical applications.
>
> **W2: "Similarly, it seems that the baselines used in the evaluation are not all fine-tuned..."**
>
> Thanks for your advice. We have evaluated the finetuning results with two baselines, Med-Flamingo and LLaVA-Med, as demonstrated in the Common Questions Q1.
>
> **W3: "Another concern about this paper is the quality of the data..."**
>
> Thanks for your valuable suggestion. We have conducted human validation for the automatically annotated data, as detailed in the Common Questions Q2. We will also verify the generated data to improve its quality before release.
>
> ## Questions
>
> **Q1: "How is the English version of the text report/caption generated for the in-house Chinese data?..."**
>
> Thanks for your question. Actually, we did not generate English version reports for the in-house Chinese data, and vice versa. We only kept the original Chinese text due to the following considerations:
>
> 1. **Aligned with real clinical scenarios**: In clinical practice, it is not necessary to generate bilingual reports because they serve people who speak respective languages. Besides, the image data distributions may vary across different populations. Separating the languages allows the model to distinguish between data distributions, thereby encouraging a better understanding of the images from different populations.
> 2. **Challenges of Translation Quality**: Creating data with both languages requires either manual translation, which is labor-intensive, or automated translation with translation models or large language models. However, due to the specialized terminology in medical reports, automated translation often yields errors or unnatural phrasing. This makes it difficult to ensure the quality of the translated data and could introduce noise into the model.
> 3. **Difference in Writing Styles**: There are differences in the writing style between Chinese and English reports. Simply using translated texts for training would affect the consistency of styles and introduce noise that would limit the model’s ability to mimic the writing style of different languages. This would lead to difficulties for doctors with different native languages in fully understanding the generated reports.
>
> **Q2: "According to the “Structure Detection” section on page 5, ..."**
>
> Thanks for your insightful question.
>
> 1. **Evaluation on the finetuned MLLM for structure detection:** We have conducted human validation for the automatically annotated data, as detailed in the Common Questions Q2.
> 2. **Comparison with specific detection models:** We applied the finetuned MLLM for structure detection instead of training a specific detection model because the MLLM allows for flexibility in customizing the target organs. Our medical report for different scans may cover different subsets of organs, and detection models with predefined targets cannot adapt dynamically to these changing objectives. Furthermore, training multiple detection models for different organs requires extra effort in dataset preparation and model deployment. By using the finetuned MLLM, we consolidate these tasks into a single model, providing a more adaptable and efficient solution.
>
> **Q3: "...the reviewer wonders how much will the other baseline medical MLLMs improve ..."**
>
> Thanks for your question. Please refer to the Common Questions Q1 for a detailed response to this question.
>
> **Q4: "...the reviewer wonders if the data will be released as well..."**
>
> Thanks for your suggestion. Our data will be released to the public.

---

> > ### Comment · Reviewer_QLEK · 2024-11-21
> > **Reply to Author's Rebuttal**
> >
> > I sincerely appreciate the author's time and effort during the rebuttal phase and it is exciting to see these new results and clarification. The authors' reply addressed most of my questions and concerns, especially on the quality of the automatically generated bounding box. I am also glad to see that the author claims that the dataset will be released to the public.
> >
> > Overall, the reviewer would like to maintain the original rating of Acceptance after reading the authors' feedback and comments from other reviewers. The reviewer believes that the major contribution of this paper, as claimed by the authors, is not only the MedRegA model but also the novel dataset that provides localized annotation along with medical images and reports. The data quality review from the expert helped to alleviate my concerns about the annotation quality. It is proved that using these data can help improve general performance according to the new experiments in the general response, even just in a few-shot style.

---

### Author Response · Authors · 2024-11-21
**General Response Part I**

We sincerely appreciate the reviewers for their time and effort in the review. We are glad that they found our proposed tasks, dataset, and model `"novel"`/`"help the development of the field"`/`"immensely useful"`/`"likely to have a significant impact on the community"` (QLEK, 5dSL, WceT) and our paper `"well written"`/`"well-organized"` (QLEK, ToYk), and agreed that our MedRegA model `"demonstrate strong performance"` and `"strong generalization"` (5dSL, WceT).

We would like to address common questions here, followed by detailed responses to each reviewer separately. We have also updated the paper with the revised parts marked in blue.

## Common Questions

**Q1: Fine-tune other baselines with our proposed dataset.**

We would like to emphasize that our core contribution is to **enhance the ability of medical MLLM to perceive and interpret regions** within the medical scan by proposing the novel MedRegInstruct dataset and the versatile MedRegA model, which remains unexplored in previous works. Specifically, we applied an automatic labeling system to curate the grounded medical reports in our dataset. Our dataset covers comprehensive region-centric tasks, aimed at encouraging the model to focus on critical areas. This significantly improves the interpretability and reliability of medical MLLM in clinical applications.

We have evaluated the finetuning results with two baselines, Med-Flamingo [1] and LLaVA-Med [2]. To save time and cost for evaluation, we use a subset of 5K samples from the proposed MedRegInstruct dataset to finetune the baseline models. This subset covers Region-to-Text Identification and Text-to-Region Detection tasks. We evaluated the models using the test split of the same subset.

For Med-Flamingo [1], a few-shot learner adapted to the medical domain, we perform a 5-shot learning with our dataset. For LLaVA-Med [2], we finetuned the model on the sub-dataset for 10 epochs. The table below shows the results.

**Results on Region-to-Text Identification Task**

| **Metric**    | **Med-Flamingo (Zero Shot)** | **Med-Flamingo (Few Shot)** | **LLaVA-Med (Base)** | **LLaVA-Med (Finetuned)** | **MedRegA** |
| ------------- | :--------------------------: | :-------------------------: | :------------------: | :-----------------------: | :---------: |
| **BLEU**      |             2.19             |            32.02            |         0.05         |           21.49           |  **59.06**  |
| **F1**        |             3.79             |            33.19            |         0.14         |           25.08           |  **59.21**  |
| **Recall**    |            11.70             |            33.40            |         1.99         |           32.23           |  **59.19**  |
| **Accuracy**  |             6.62             |            16.87            |         0.91         |           31.93           |  **51.72**  |
| **BertScore** |            50.36             |            71.53            |        32.18         |           58.21           |  **82.49**  |

**Results on Text-to-Region Detection Task**

| **Metric**                     | **Med-Flamingo (Zero Shot)** | **Med-Flamingo (Few Shot)** | **LLaVA-Med (Base)** | **LLaVA-Med (Finetuned)** | **MedRegA** |
| ------------------------------ | :--------------------------: | :-------------------------: | :------------------: | :-----------------------: | :---------: |
| **Region-Level F1**            |             N/A              |            18.36            |         N/A          |           19.68           |  **56.08**  |
| **Alignment F1** |             N/A              |            20.65            |         N/A          |           19.53           |  **52.52**  |
| **IoU**                        |             N/A              |            19.52            |         N/A          |           25.27           |  **47.43**  |

In the table, 'N/A' indicates that the model cannot generate valid outputs

From the results, we can observe that both Med-Flamingo [1] and LLaVA-Med [2] can improve on the regional tasks even in a few-shot setting or fine-tuning with a small subset in our proposed dataset. For example, they achieve an increase of 29.4% and 24.94% respectively in the F1 score on Region-to-Text Identification task. This proves the effectiveness of our dataset in extending regional knowledge.

We have updated the results and analysis above in **Appendix C.3**.

[1] Moor, Michael, et al. "Med-flamingo: a multimodal medical few-shot learner." Machine Learning for Health (ML4H). PMLR, 2023.

[2] Li, Chunyuan, et al. "Llava-med: Training a large language-and-vision assistant for biomedicine in one day." Advances in Neural Information Processing Systems 36 (2024).

---

> ### Author Response · Authors · 2024-11-21
> **General Response Part II**
>
> **Q2: Human validation of the quality of automatically generated annotations.**
>
> To construct report grounding dataset for our in-house data, we implemented two automated processes: (1) **Report Refinement**: we employed InternLM to segment each report into detailed descriptions for each organ; (2) **Structure Detection**: We first finetuned an MLLM based on the Region-Text dataset to enable it to generate regions based on organ names. Then, we input the collected images into the finetuned MLLM, and prompted it with the organs mentioned in the reports to obtain the corresponding bounding boxes.
>
> To evaluate the quality of the automatically annotated part of our dataset, we randomly selected 50 samples and asked 2 experts to create manual labels for comparison.
>
> 1. **Human Validation of Report Refinement**
>
>    Report segmentation is a relatively straightforward task for a large language model, as it does not require generating new information but simply categorizing existing content based on the organs described. The sentence-level accuracy is **93.33%**, where **2.67%** of the sentences are missing and **4%** of them are wrongly classified.
>
> 2. **Human Validation of Structure Detection**
>
>    Compared with the human annotation, the accuracy of generated bounding boxes is **72%**, and the IoU score is **55%**. We also conducted a visual evaluation and found that although most bounding boxes are slightly larger, they can still encompass the target region. This is sufficient for our approach since we only require localization rather than a tight and accurate bounding box.
>
> We have included the human validation in **Section 3.2** to provide a quantitive assessment of the data quality.

---

### Meta-Review · Area_Chair_5pPM · 2024-12-19

**Metareview:**

This paper developed MedRegA, a Region-Aware medical Multimodal Large Language Model (MLLM) designed to enhance interpretability and interactivity by incorporating anatomical region awareness into medical vision-language tasks across multiple modalities. By leveraging a newly constructed dataset, MedRegInstruct, and enabling both image-level and region-level tasks, MedRegA achieves state-of-the-art performance in bilingual medical applications, including visual question answering, report generation, and image classification.

During the initial review, reviewers praised that this paper introduces the novel MedRegInstruct dataset and the region-aware MedRegA model with an innovative Regional Chain-of-Thought approach, achieving state-of-the-art performance and addressing critical gaps in fine-grained region-specific vision-language tasks.  On the other hand, they also found that the main weaknesses of this paper are the limited novelty of the MedRegA model, as it heavily relies on existing architectures like InternVL with minor modifications in training schemes and prompts; concerns about fairness in baseline comparisons, as many baselines were not fine-tuned on the proposed dataset; and the lack of human validation to ensure the quality of the dataset’s automated annotations.

During the discussion, the authors provided additional results of finetuning two baselines, Med-Flamingo and LLaVA-Med, and some clarifications about human evaluation. They also provided new results comparing report generation models on IU-Xray and MIMIC-CXR datasets. The reviewers collectively appreciated the authors' detailed rebuttal, clarifications, and additional results, particularly regarding the quality and impact of the MedRegInstruct dataset and the MedRegA model. The dataset's potential to benefit the research community was widely recognized, leading to improved ratings from 3 reviewers, with overall support for acceptance.

The reviewers have reached a consensus, leaning toward accepting the paper. This AC agrees with their assessment of the proposed dataset's significant potential contribution and impact on the research community, which supports the paper's acceptance. While the MedRegA model is not entirely novel, it establishes a solid performance baseline for the dataset. However, this AC strongly encourages the authors to enhance and clarify their discussion of human validation and dataset quality in the final version of the paper. For instance, statements such as “We also conducted a visual evaluation and found that although most bounding boxes are slightly larger, they can still encompass the target region” remain vague and could benefit from more specific details and justification.

**Additional Comments On Reviewer Discussion:**

Reviewers found that the main weaknesses of this paper are the limited novelty of the MedRegA model, as it heavily relies on existing architectures like InternVL with minor modifications in training schemes and prompts; concerns about fairness in baseline comparisons, as many baselines were not fine-tuned on the proposed dataset; and the lack of human validation to ensure the quality of the dataset’s automated annotations.

During the discussion, the authors provided additional results of finetuning two baselines, Med-Flamingo and LLaVA-Med, and some clarifications about human validation. They also provided new results comparing report generation models on IU-Xray and MIMIC-CXR datasets. The reviewers collectively appreciated the authors' detailed rebuttal, clarifications, and additional results, particularly regarding the quality and impact of the MedRegInstruct dataset and the MedRegA model. Concerns about annotation quality were alleviated by expert reviews, and new experiments demonstrated the dataset's utility in improving performance. The dataset's potential to benefit the research community was widely recognized, leading to improved ratings from 3 reviewers (5dSL, ToYk, and WceT), with overall support for acceptance.

---

### Decision · Program_Chairs · 2025-01-22

Accept (Poster)